# Long COVID risk and pre-COVID vaccination in an EHR-based cohort study from the RECOVER program

M. Daniel Brannock [1,15] ✉, Robert F. Chew [1,15], Alexander J. Preiss [1,15], Emily C. Hadley [1,15], Signe Redfield [2], Julie A. McMurry [3], Peter J. Leese [4], Andrew T. Girvin [5], Miles Crosskey [6], Andrea G. Zhou [7], Richard A. Moffitt [8,9], Michele Jonsson Funk [4], Emily R. Pfaff [4], Melissa A. Haendel [3], Christopher G. Chute [10], N3C* & RECOVER Consortia*

Long COVID, or complications arising from COVID-19 weeks after infection, has become a central concern for public health experts. The United States National Institutes of Health founded the RECOVER initiative to better understand long COVID. We used electronic health records available through the National COVID Cohort Collaborative to characterize the association between SARS-CoV-2 vaccination and long COVID diagnosis. Among patients with a COVID-19 infection between August 1, 2021 and January 31, 2022, we defined two cohorts using distinct definitions of long COVID—a clinical diagnosis ($n = 47,404$) or a previously described computational phenotype ($n = 198,514$)—to compare unvaccinated individuals to those with a complete vaccine series prior to infection. Evidence of long COVID was monitored through June or July of 2022, depending on patients' data availability. We found that vaccination was consistently associated with lower odds and rates of long COVID clinical diagnosis and high-confidence computationally derived diagnosis after adjusting for sex, demographics, and medical history.

The SARS-CoV-2 virus, and the COVID-19 pandemic it effected, hardly needs introducing more than 2 years after the World Health Organization (WHO) first announced evidence of human-to-human transmission in January of 2020[1]. As of this writing, the WHO states there have been 626 million confirmed cases and more than 6 million deaths attributed to COVID-19 worldwide[2]. Post-acute sequelae of SARS-CoV-2 infection (PASC) have been widely reported and can include any complication resulting from SARS-CoV-2 infection weeks or months after infection[3–5]. Long COVID is a single diagnosis that encapsulates a broad array of symptoms attributed to PASC. The WHO used a Delphi method to create a clinical definition of long COVID that includes both clinically observed features as well as patient-reported features[6]. Long COVID is a multi-system disease, characterized by diverse features such as dyspnea, chest pain, fatigue, cognitive impairment, deep vein thrombosis, gastrointestinal dysfunction, etc.[7,8]. Numerous efforts to define long COVID using electronic health record (EHR) data exist, with the goals of supporting public health surveillance and research[9]. However, a gold standard definition of long COVID has been elusive. We have previously provisioned the first machine learning EHR-based long COVID definition (a Computable Phenotype, or CP model), which

[1]RTI International, Durham, NC, USA. [2]Patient-Led Research Collaborative, Pomfret, MD, USA. [3]University of Colorado Anschutz Medical Campus, Denver, CO, USA. [4]University of North Carolina at Chapel Hill, Chapel Hill, NC, USA. [5]Palantir Technologies, Denver, CO, USA. [6]CoVar Applied Technologies, Durham, NC, USA. [7]iTHRIV, University of Virginia, Charlottesville, VA, USA. [8]Department of Biomedical Informatics, Stony Brook University, Stony Brook, NY, USA. [9]Departments of Biomedical Informatics and Hematology and Medical Ontology, Emory University, Atlanta, GA, USA. [10]Schools of Medicine, Public Health, and Nursing, Johns Hopkins University, Baltimore, MD, USA. [15]These authors contributed equally: M. Daniel Brannock, Robert F. Chew, Alexander J. Preiss, Emily C. Hadley. *Lists of authors and their affiliations appear at the end of the paper. ✉e-mail: mbrannock@rti.org

**Table 1 | Model-based cohort patient demographics**

| Variable | Overall (N = 198514) | Fully vaccinated (N = 86,248) | Unvaccinated (N = 112,266) |
|---|---|---|---|
| Mean age | 47.23 (100.0) | 52.54 (100.0) | 43.16 (100.0) |
| **Age at COVID index date** | | | |
| 18–24 | 20603 (10.4) | 4841 (5.6) | 15762 (14.0) |
| 25–34 | 36515 (18.4) | 11197 (13.0) | 25318 (22.6) |
| 35–49 | 53548 (27.0) | 21355 (24.8) | 32193 (28.7) |
| 50–64 | 50644 (25.5) | 25118 (29.1) | 25526 (22.7) |
| 65+ | 37204 (18.7) | 23737 (27.5) | 13467 (12.0) |
| **Sex** | | | |
| Female | 128269 (64.6) | 55998 (64.9) | 72271 (64.4) |
| Male | 70245 (35.4) | 30250 (35.1) | 39995 (35.6) |
| **Race/Ethnicity** | | | |
| Asian non-Hispanic | 2651 (1.3) | 2105 (2.4) | 546 (0.5) |
| Black or African American non-Hispanic | 26397 (13.3) | 10173 (11.8) | 16224 (14.5) |
| Hispanic or Latino any race | 18748 (9.4) | 9773 (11.3) | 8975 (8.0) |
| Pacific Islander non-Hispanic | 266 (0.1) | 151 (0.2) | 115 (0.1) |
| Other non-Hispanic | 4219 (2.1) | 1460 (1.7) | 2759 (2.5) |
| Unknown | 3554 (1.8) | 1985 (2.3) | 1569 (1.4) |
| White non-Hispanic | 142679 (71.9) | 60601 (70.3) | 82078 (73.1) |
| **Data partner** | | | |
| Partner A | 9471 (4.8) | 5816 (6.7) | 3655 (3.3) |
| Partner B | 1757 (0.9) | 1166 (1.4) | 591 (0.5) |
| Partner C | 3424 (1.7) | 2096 (2.4) | 1328 (1.2) |
| Partner D | 1139 (0.6) | 736 (0.9) | 403 (0.4) |
| Partner E | 2721 (1.4) | 2080 (2.4) | 641 (0.6) |
| Partner F | 27179 (13.7) | 14714 (17.1) | 12465 (11.1) |
| Partner G | 6081 (3.1) | 3971 (4.6) | 2110 (1.9) |
| Partner H | 2123 (1.1) | 984 (1.1) | 1139 (1.0) |
| Partner I | 2220 (1.1) | 1430 (1.7) | 790 (0.7) |
| Partner J | 25794 (13.0) | 12677 (14.7) | 13117 (11.7) |
| Partner K | 116605 (58.7) | 40578 (47.0) | 76027 (67.7) |

**Table 2 | Model-based cohort medical characteristics**

| Variable | Overall (N = 198,514) | Fully vaccinated (N = 86,248) | Unvaccinated (N = 112,266) |
|---|---|---|---|
| **Month of COVID index date** | | | |
| August 2021 | 48056 (24.2) | 15029 (17.4) | 33027 (29.4) |
| September 2021 | 45278 (22.8) | 15831 (18.4) | 29447 (26.2) |
| October 2021 | 22832 (11.5) | 9282 (10.8) | 13550 (12.1) |
| November 2021 | 23999 (12.1) | 10143 (11.8) | 13856 (12.3) |
| December 2021 | 23248 (11.7) | 12083 (14.0) | 11165 (9.9) |
| January 2022 | 35101 (17.7) | 23880 (27.7) | 11221 (10.0) |
| **Health status** | | | |
| Immunocompromised | 2139 (1.1) | 1537 (1.8) | 602 (0.5) |
| Diabetes (Complicated) | 15152 (7.6) | 9137 (10.6) | 6015 (5.4) |
| Diabetes (Uncomplicated) | 30383 (15.3) | 16912 (19.6) | 13471 (12.0) |
| Kidney disease | 13145 (6.6) | 7973 (9.2) | 5172 (4.6) |
| Acute kidney injury | 7570 (3.8) | 4248 (4.9) | 3322 (3.0) |
| Chronic lung disease | 26487 (13.3) | 13827 (16.0) | 12660 (11.3) |
| Tobacco smoker | 8301 (4.2) | 3149 (3.7) | 5152 (4.6) |
| Heart failure | 8241 (4.2) | 4893 (5.7) | 3348 (3.0) |
| Myocardial infarction | 4924 (2.5) | 2702 (3.1) | 2222 (2.0) |
| **Weeks since last vaccination** | | | |
| <2 | 2662 (1.3) | 2662 (3.1) | 0 (0.0) |
| 2–4 | 1947 (1.0) | 1947 (2.3) | 0 (0.0) |
| 5–9 | 5596 (2.8) | 5596 (6.5) | 0 (0.0) |
| 10–14 | 7404 (3.7) | 7404 (8.6) | 0 (0.0) |
| 15–19 | 10011 (5.0) | 10011 (11.6) | 0 (0.0) |
| 20–24 | 12566 (6.3) | 12566 (14.6) | 0 (0.0) |
| ≥25 | 46062 (23.2) | 46062 (53.4) | 0 (0.0) |
| Unvaccinated | 112266 (56.6) | 0 (0.0) | 112266 (100.0) |

diagnosis or computable phenotype[10]; previous studies have relied on surveys or the occurrence of one or two symptoms consistent with long or acute COVID. Ours is also the largest study to leverage time-to-event modeling or control for differences in the vaccinated and unvaccinated populations.

## Results

Patients with a COVID-19 infection between August 1, 2021 and January 31, 2022 were split into two cohorts. In a clinic-based cohort of 47,404 individuals, 695 (1.5%) received a clinical diagnosis of long COVID and 26,354 (55.6%) were fully vaccinated (Supplementary Table 1 shows the number of individuals with each of the long COVID diagnoses; a single person can receive multiple diagnoses). In a model-based cohort of 198,514 individuals, 3391 (1.7%) had a computational phenotype[10] (CP) score above a threshold of 0.9 and were labeled as having long COVID; 86,248 (43.4%) were fully vaccinated. All available EHRs beginning 45 days after COVID-19 infection were used to establish evidence of long COVID. The end of patients' follow-up periods varied depending on the cadence of their healthcare facility's data contributions and ranged from June 10, 2022 to August 1, 2022. The minimum observed follow-up period between COVID-19 infection and the end of an individual's data availability was 164 days. Distributions of the length of the follow-up period for both cohorts are shown in Supplementary Table 2. Full summaries of patient characteristics for both cohorts are shown in Tables 1–4. Unadjusted cross-tabulations of vaccination status and long COVID diagnosis are shown in Table 5.

### Statistical analysis

Inverse probability of treatment weighting (IPTW) was applied to logistic regression and Cox proportional hazards models for both

leveraged the long COVID ICD-10 code U09.9 as well as visits to long COVID specialty clinics to train a classifier to identify putative long COVID Patients[10]; as well as later work by others[11].

The National Institutes of Health (NIH) created the RECOVER initiative to address the uncertainty surrounding long COVID by coordinating research across hundreds of researchers and more than 30 institutions[12]. The National COVID Cohort Collaborative (N3C)[13], sponsored by NIH's National Center for Advancing Translational Sciences, provides access to harmonized electronic health records through the N3C Data Enclave. More than 75 sites have contributed longitudinal data for over 15.5 million patients with a confirmed SARS-CoV-2 infection, COVID-19 symptoms, or their matched controls.

Vaccines have been shown to be safe and effective at dramatically reducing the risk of severe COVID-19[14,15]. However, their impact on long COVID is less understood, with most studies indicating a significant protective effect[16–19] while others reported mixed or no effects[20], or even an anti-protective effect[21]. While some have studied the impact of administering vaccines after the onset of PASC[22–24], we attempt to address ambiguity around the association between pre-COVID-19 vaccination and eventual long COVID diagnosis. To our knowledge, we are the first to consider vaccination with long COVID as a clinical

**Table 3 | Clinic-based cohort patient demographics**

| Variable | Overall (*N* = 47,404) | Fully vaccinated (*N* = 26,354) | Unvaccinated (*N* = 21,050) |
|---|---|---|---|
| Mean age | 48.19 (100.0) | 51.01 (100.0) | 44.67 (100.0) |
| **Age at COVID index date** | | | |
| 18–24 | 4483 (9.5) | 1731 (6.6) | 2752 (13.1) |
| 25–34 | 8486 (17.9) | 4037 (15.3) | 4449 (21.1) |
| 35–49 | 12334 (26.0) | 6681 (25.4) | 5653 (26.9) |
| 50–64 | 12268 (25.9) | 7179 (27.2) | 5089 (24.2) |
| 65+ | 9833 (20.7) | 6726 (25.5) | 3107 (14.8) |
| **Sex** | | | |
| Female | 30819 (65.0) | 17356 (65.9) | 13463 (64.0) |
| Male | 16585 (35.0) | 8998 (34.1) | 7587 (36.0) |
| **Race/Ethnicity** | | | |
| Asian non-Hispanic | 855 (1.8) | 666 (2.5) | 189 (0.9) |
| Black or African American non-Hispanic | 10480 (22.1) | 5283 (20.0) | 5197 (24.7) |
| Hispanic or Latino any race | 3053 (6.4) | 1625 (6.2) | 1428 (6.8) |
| Pacific Islander non-Hispanic | 73 (0.2) | 37 (0.1) | 36 (0.2) |
| Other non-Hispanic | 1385 (2.9) | 454 (1.7) | 931 (4.4) |
| Unknown | 1166 (2.5) | 623 (2.4) | 543 (2.6) |
| White non-Hispanic | 30392 (64.1) | 17666 (67.0) | 12726 (60.5) |
| **Data partner** | | | |
| Partner A | 9311 (19.6) | 5738 (21.8) | 3573 (17.0) |
| Partner B | | | |
| Partner C | 3389 (7.1) | 2077 (7.9) | 1312 (6.2) |
| Partner D | 1133 (2.4) | 734 (2.8) | 399 (1.9) |
| Partner E | | | |
| Partner F | | | |
| Partner G | 5993 (12.6) | 3912 (14.8) | 2081 (9.9) |
| Partner H | | | |
| Partner I | 2156 (4.5) | 1396 (5.3) | 760 (3.6) |
| Partner J | 25422 (53.6) | 12497 (47.4) | 12925 (61.4) |
| Partner K | | | |

**Table 4 | Clinic-based cohort medical characteristics**

| Variable | Overall (*N* = 47,404) | Fully vaccinated (*N* = 26,354) | Unvaccinated (*N* = 21,050) |
|---|---|---|---|
| **Age at COVID index date** | | | |
| 18–24 | 4483 (9.5) | 1731 (6.6) | 2752 (13.1) |
| 25–34 | 8486 (17.9) | 4037 (15.3) | 4449 (21.1) |
| 35–49 | 12334 (26.0) | 6681 (25.4) | 5653 (26.9) |
| 50–64 | 12268 (25.9) | 7179 (27.2) | 5089 (24.2) |
| 65+ | 9833 (20.7) | 6726 (25.5) | 3107 (14.8) |
| **Month of COVID index date** | | | |
| August 2021 | 7361 (15.5) | 2848 (10.8) | 4513 (21.4) |
| September 2021 | 7221 (15.2) | 3120 (11.8) | 4101 (19.5) |
| October 2021 | 3448 (7.3) | 1687 (6.4) | 1761 (8.4) |
| November 2021 | 2920 (6.2) | 1497 (5.7) | 1423 (6.8) |
| December 2021 | 7496 (15.8) | 4485 (17.0) | 3011 (14.3) |
| January 2022 | 18958 (40.0) | 12717 (48.3) | 6241 (29.6) |
| **Health status** | | | |
| Immunocompromised | 1284 (2.7) | 1000 (3.8) | 284 (1.3) |
| Diabetes (Complicated) | 5458 (11.5) | 3633 (13.8) | 1825 (8.7) |
| Diabetes (Uncomplicated) | 9766 (20.6) | 6070 (23.0) | 3696 (17.6) |
| Kidney disease | 5214 (11.0) | 3430 (13.0) | 1784 (8.5) |
| Acute kidney injury | 3529 (7.4) | 2164 (8.2) | 1365 (6.5) |
| Chronic lung disease | 10175 (21.5) | 6036 (22.9) | 4139 (19.7) |
| Tobacco smoker | 5012 (10.6) | 2026 (7.7) | 2986 (14.2) |
| Heart failure | 3419 (7.2) | 2184 (8.3) | 1235 (5.9) |
| Myocardial infarction | 2204 (4.6) | 1296 (4.9) | 908 (4.3) |
| **Weeks since last vaccination** | | | |
| <2 | 851 (1.8) | 851 (3.2) | 0 (0.0) |
| 2–4 | 709 (1.5) | 709 (2.7) | 0 (0.0) |
| 5–9 | 2151 (4.5) | 2151 (8.2) | 0 (0.0) |
| 10–14 | 3289 (6.9) | 3289 (12.5) | 0 (0.0) |
| 15–19 | 3201 (6.8) | 3201 (12.1) | 0 (0.0) |
| 20–24 | 2916 (6.2) | 2916 (11.1) | 0 (0.0) |
| ≥25 | 13237 (27.9) | 13237 (50.2) | 0 (0.0) |
| Unvaccinated | 21050 (44.4) | 0 (0.0) | 21050 (100.0) |

cohorts to account for known confounders between vaccination propensity and risk of long COVID. The result is four adjusted estimates, which showed consistent protective associations between vaccination and long COVID diagnosis and are reported in Table 6. The full tables of model coefficients are provided as Supplementary Tables 3–6. Unadjusted estimates are also reported in Table 6, which do not exhibit the same association. The IPTW-adjusted Kaplan–Meier curves for the model-based and clinic-based outcomes are shown in Fig. 1.

Key results of the sensitivity analyses are summarized in Fig. 2. Adjusted and unadjusted estimates were evaluated across multiple CP score thresholds, and by including or excluding covariates in addition to vaccination status. The association between vaccination and long COVID was robust to excluding either IPTW-adjustment or non-vaccination covariates, but not both. While not relevant in the clinic-based outcome, the association in the model-based outcome was not robust to the varying CP thresholds, with lower thresholds resulting in a progressively weaker protective association. In the proportional hazards models, an additional analysis determined that the estimates were not sensitive to whether or not post-COVID-19 vaccination events are censoring events (uncensored points are not pictured in Fig. 2 as

they closely overlap the censored points). The remaining sensitivity analysis results are shown in Supplementary Table 7. In the clinic-based cohort, an analysis showed the results to be robust to using only the most specific ICD-10 code (U09.9) to label long COVID. Including only sites with the most complete vaccine reporting (with recorded vaccine ratios of at least 89%) resulted in associations similar to or stronger than the four primary associations. Censoring patients in the clinic-based analysis after their last recorded healthcare visit and eliminating the requirement for a post-COVID-19 visit resulted in a slightly stronger, but not significantly different, association as in our primary results.

The subanalysis did not offer robust evidence that the association between vaccination and long COVID diagnosis is dependent on the time between vaccination and acute COVID-19 onset. The full tables of subanalysis coefficients, including for indicators of vaccination timing, are shown in Supplementary Tables 8–11.

After IPTW-adjustment, all covariates were well-balanced (Supplementary Figs. 1, 2 illustrate the standardized differences in covariates in both cohorts). Logistic regression diagnostics did not indicate any overly influential observations. Observations with large residuals tended to have low leverage and vice versa. In the model-based analysis, the greatest Cook's distance was <0.01 and the greatest absolute DFBETA for vaccination status was 0.07. In the clinic-based analysis,

**Table 5 | Long COVID by vaccination status: unadjusted counts**

| | Model-based cohort | | | Clinic-based cohort | | |
|---|---|---|---|---|---|---|
| | Overall | With long COVID | Without long COVID | Overall | With long COVID | Without long COVID |
| Fully vaccinated | 86,248 (100%) | 1506 (1.7%) | 84,742 (98.3%) | 26,354 (100%) | 346 (1.3%) | 26,008 (98.7%) |
| Unvaccinated | 112,266 (100%) | 1885 (1.7%) | 110,381 (98.3%) | 21,050 (100%) | 349 (1.7%) | 20,701 (98.3%) |

An individual must have received two doses of BNT162b2 or mRNA-1273 or a single dose of Ad26.COV2.S at least 2 weeks prior to COVID index to be fully vaccinated. Unvaccinated individuals received no doses prior to the COVID index.

**Table 6 | Long COVID by vaccination status: measures of association**

| | Logistic regression OR[a] (95% CI) | Proportional hazards HR[b] (95% CI) |
|---|---|---|
| **IPTW-adjusted** | | |
| Model-based outcome | 0.70 (0.65, 0.75) | 0.63 (0.57, 0.69) |
| Clinic-based outcome | 0.70 (0.60, 0.81) | 0.67 (0.56, 0.79) |
| **Unadjusted** | | |
| Model-based outcome | 1.04 (0.97, 1.11) | 1.00 (0.91, 1.10) |
| Clinic-based outcome | 0.79 (0.68, 0.92) | 0.79 (0.66, 0.95) |

[a]*OR* Odds ratio
[b]*HR* Hazard ratio

the greatest Cook's distance was 0.01 and the greatest absolute DFBETA for vaccination status was 0.09. In the model-based analysis, five patients had stabilized inverse probability of treatment weights above 20 (max of 32); excluding these patients did not impact vaccination coefficients at the precision reported here. The maximum weight in the clinic-based analysis was ten.

## Discussion

Our four analyses yielded consistent results. We see protective associations of vaccination with long COVID diagnosis in both logistic and time-to-event models, and in both clinic-based and model-based outcomes. While these findings are similar to those of other large observational studies[16–19], previous sources have only looked for evidence of COVID-associated symptoms as evidence of long COVID. A major finding of our analysis is that the protective association remains consistent in results requiring a clinical diagnosis, and among those who contracted COVID-19 in a later period that includes Omicron infections.

The use of a clinical diagnosis resulted in a significantly lower long COVID prevalence in our study (less than 2% in both cohorts) than studies based on long COVID symptoms, which have reported prevalences between 8 and 38%, depending on which and how many symptoms were required[17–19]. However, both of our cohorts are large, and the use of a CP allowed us to expand our sample from six to eleven sites and 47,404 to 198,514 COVID-positive patients, providing a sufficient sample of strictly defined long COVID diagnoses. Due to the underdiagnosis of long COVID in a clinical setting, our conclusions are limited to associations with diagnosis and not with long COVID onset more generally.

Interestingly, the protective association of vaccination with long COVID diagnosis is weaker or reversed in the unadjusted coefficients and cross-tabulations (Table 6 and Fig. 2). Several features that are associated with a higher likelihood of long COVID (coefficients in Supplementary Tables 3–6) are also associated with a higher likelihood of vaccination (coefficients in Supplementary Tables 12, 13). The most significant is age: Supplementary Table 14 shows how older adults are both more likely to be vaccinated and more likely to contract long COVID in comparison to younger adults. Failing to account for the substantial differences between individuals who were and were not vaccinated prior to COVID-19 could lead one to inaccurately conclude that vaccination is harmful.

The sensitivity analysis presents other instructive complexities. Reducing the CP score threshold lowers the amount of evidence required to denote someone as having long COVID; it also moderates the protective association of vaccination with long COVID (key results in Fig. 2, full range of thresholds in Supplementary Fig. 3). We expect that including healthy adults in the long COVID, population would dilute the observed association, but individuals with a CP score between 0.6 and 0.9 are not entirely healthy—they have some evidence of long COVID. In fact, our sample's long COVID incidence rate at lower thresholds is closer to long COVID incidence rates reported elsewhere (although the true incidence rate of long COVID is unknown). This suggests a hypothesis that vaccination may be more effective at preventing clinically diagnosed long COVID than undiagnosed long COVID. More research is needed to determine the differences between high confidence and clinically diagnosed long COVID cases compared to low confidence and undiagnosed cases. If they are more severe, then our results could suggest that vaccination is associated with reduced severity of long COVID symptoms.

Healthcare utilization is one of the most important features of the CP model[10]. If fully vaccinated patients are more likely to utilize the healthcare system, the CP model's marginal predictions may be assigning more fully vaccinated individuals to long COVID because they are more likely to interact with the healthcare system, depressing the observed benefit of vaccination. A known challenge of analyzing EHR data is that they tend to provide more information on individuals who regularly utilize healthcare systems[25], though we attempt to control for this by requiring multiple recorded encounters outside of COVID-19 for inclusion in the study.

Our use of long COVID diagnosis and a computable phenotype as outcomes differentiate this study from others[17,18,20], which measure the association between vaccination and a curated list of long COVID symptoms. Each approach has its strengths. Our clinical outcomes reduce measurement error due to false positives (e.g., long COVID symptoms caused by something other than long COVID). However, other studies show that long COVID symptoms differ in their relationship with vaccination. Our outcomes obscure such variation. We conclude that it is beneficial to study this relationship from both perspectives.

Vaccination reduces the risk of developing COVID-19 for a period of time after vaccination[14,15], offering one mechanism for preventing long COVID. However, there is evidence that widely circulated vaccines are less effective against now-dominant Omicron than earlier SARS-CoV-2 variants[26–28], increasing interest in whether or not vaccination reduces the risk of long COVID in breakthrough infections. That is the aim of this study, in which all eligible patients had a COVID-19 diagnosis. As a result, we are excluding any effect due to vaccination's primary prevention of COVID-19 in the first place that is present in the general population.

Several studies conclude that the protective effect of vaccination on acute COVID-19 infection severity wanes over time[27,29], but we are unaware of any studies making the same claim for long COVID. As can be seen in Supplementary Tables 8–11, the subanalysis incorporating time between vaccination and acute COVID-19 does not offer any evidence that the association between vaccination and long COVID diagnosis changes over time. The reference level for Weeks Since Last Vaccination is those who received their last vaccine dose at least

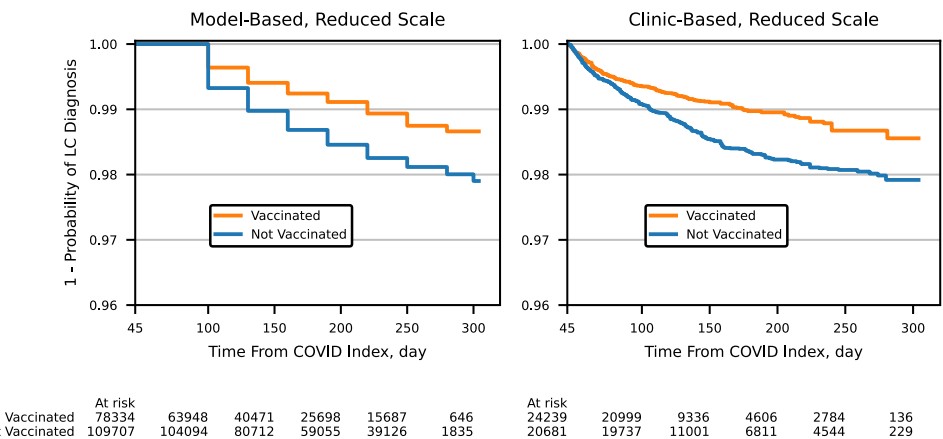

**Fig. 1 | IPTW-adjusted Kaplan–Meier curves.** Our definition of long COVID (LC) can only be observed at least 45 days after index; time from the COVID index therefore starts at 45. Long COVID events can only be observed for the model-based outcome in 30-day increments, resulting in the observed stair-step structure. A reduced vertical axis scale is used to highlight the differentiation between the vaccinated and unvaccinated curves.

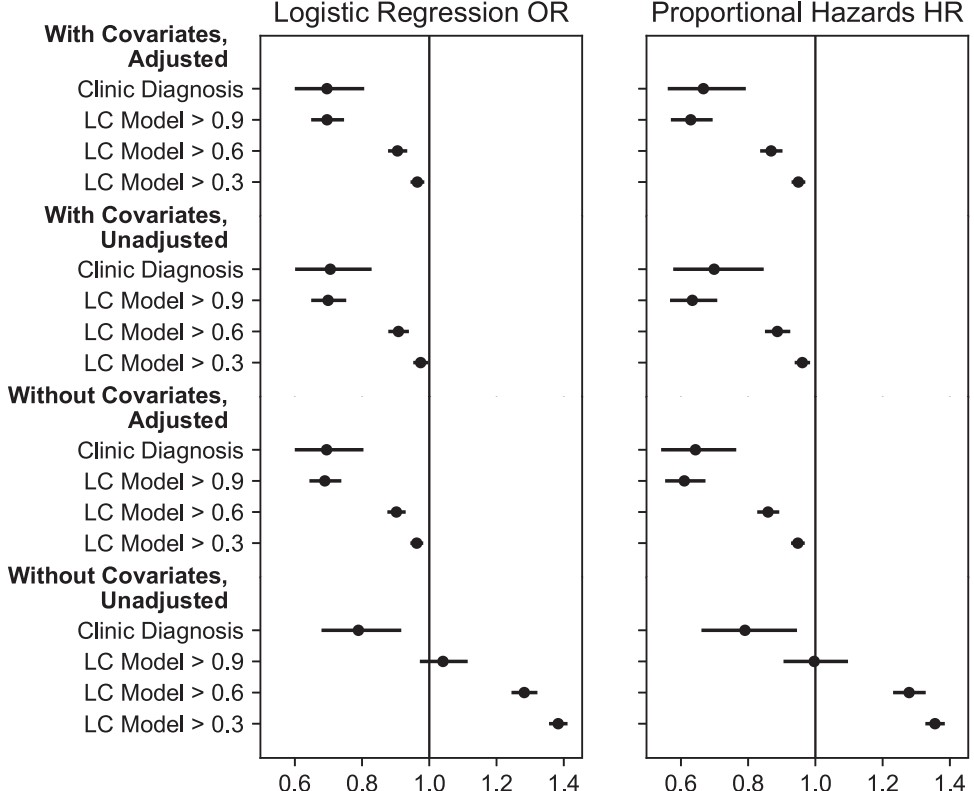

**Fig. 2 | Sensitivity analysis of vaccination associations.** Odds ratios (OR) are shown for logistic regression (LR), hazard ratios (HR) are shown for proportional hazards (PH). Point estimates are from models built using the full cohorts and are shown with 95% confidence intervals derived from 200 bootstrap samples. The vertical line at 1.0 represents no association. The clinic diagnosis points (*n* = 47,404 individuals) are using the clinic-based outcome, the long COVID (LC) model points ($n$ = 198,514 individuals) represent different thresholds of the computational phenotype model to label LC. Higher thresholds represent higher confidence in an LC phenotype. With or without covariates refers to the presence or absence of non-vaccination predictors in the outcome models. Adjusted or unadjusted refers to the presence or absence of IPTW weighting.

25 weeks prior to their COVID-19 infection. Negative coefficients for the modeled indicators suggest stronger protective associations. Three models present statistical significance (alpha = 0.05) for at least one indicator, indicating a significant difference between that level and those vaccinated 25+ weeks prior to COVID-19, but results are not consistent between models. Contrary to intuition and previously reported results with acute COVID-19, those vaccinated at least 25 weeks prior to COVID-19 are among the least likely to be diagnosed with long COVID across the four models. We do not present this as

evidence that the benefits of vaccination with respect to long COVID do not wane. Caution should be used when interpreting conditional coefficients and investigating the time between vaccination and COVID-19 was not a primary focus in this study[30].

IPTW is often used to estimate causal effects from observational data and is employed here to provide more robust associations. However, we do not interpret these results as causal effects. This is for two reasons: (1) we are unwilling to assume that there are no unmeasured confounders in our treatment model and (2) our causal model

includes several latent variables, which obstruct the estimation of treatment effects through covariate adjustment. We explore each reason in the Supplementary Discussion and provide a directed acyclic graph of confounders in Supplementary Fig. 4.

Our study is limited by its reliance on EHRs and other factors. Those who choose to not seek healthcare, or are unable to do so, are not represented in EHRs. This could be particularly problematic among long COVID patients, who may lack the energy or resources required to receive a clinical diagnosis, or whose providers may not be familiar enough with long COVID symptoms to make a diagnosis. If vaccinated long COVID patients are less likely to be clinically diagnosed than unvaccinated long COVID patients, then our estimate of the association between vaccination and long COVID diagnosis will overstate the association between vaccination and long COVID onset. Furthermore, we had previously identified a heterogeneous set of features that were differentially present in clinical observations versus patient-reported symptoms[9]. This agreed with the WHO suggestion that a definition of long COVID must necessarily include both clinician and patient-reported features - which are not commonly available in the EHR.

We are forced to assume that those without a recorded condition or symptom do not exhibit it, including our exposure (vaccination), our outcome (long COVID diagnosis), and potentially unrecorded reinfections of COVID-19. We take two steps to mitigate the risk of unrecorded records relevant to the outcome: (1) we require that all participants in the study had established care at the partner facility prior to COVID-19 infection, as evidenced by two healthcare visits in the year prior, and (2) we require that all participants in the study were seen at the partner facility at least 120 days after COVID-19 infection. Our utilization-related inclusion criteria result in a cohort that is disproportionately female, which may be due in part to females being higher healthcare utilizers than males on average[31–33]. We account for confounding due to sex by including sex as a covariate in treatment weighting and in all primary models. The utilization criteria result in a significantly smaller cohort and biases the sample towards high utilizers and those with hospitalizations. However, it remains sufficiently large for analysis and has a lower risk that long COVID will go undiagnosed, as patients were active users of the partner facility both before and four months after COVID-19 infection. In the clinic-based cohort, there is an additional requirement that the facilities have a track record of diagnosing long COVID (though the variation between doctors remains).

A sensitivity analysis that censors individuals after their last healthcare visit (rather than at the end of the study period) yields an association similar to our four primary results (Supplementary Table 7). Censoring is not possible in logistic regression models, but allows the proportional hazards model to relax the assumption that individuals that established care at a facility continue to use that facility after their last recorded visit. For this analysis, we did not require a recorded visit after the acute COVID-19 infection, but individuals remained ineligible for long COVID designation until 45 days after infection.

Our cohort is further refined by our requirement that the partner facility have reasonably high recorded vaccine ratios, as defined in our Methods. Most facilities fail to achieve recorded vaccine ratios greater than 66%, as they are not the primary provider of vaccinations in their community, do not link to their state's vaccine registry, or do not consistently record the vaccinations they provide in the EHR. We do not use a facility's vaccination rate as an individual characteristic in our models, but rather as a facility-wide inclusion criterion. By limiting to partner facilities with a high vaccination rate, as with our utilization criteria, we refine our cohort to be smaller but more data-rich.

We strictly define our study cohort to minimize the underreporting of vaccination and long COVID, though we acknowledge that it is not entirely resolved. Our sensitivity analysis using only sites with the highest recorded vaccine ratios (≥89%) offers some evidence that incomplete vaccine records result in a conservative estimate. The

cohorts are small (the model-based cohort has 10,122 patients; the clinic-based cohort has 5545), resulting in wide confidence intervals that include the primary estimates for every model (Supplementary Table 7). However, the mean estimated associations are stronger than our primary results in three of the four models and remain significant in all four models with 95% confidence. We conclude that our primary estimates are likely conservative, but our primary result—that pre-COVID-19 vaccination is associated with a reduced risk of long COVID diagnosis—is not threatened.

The confidence intervals around the CP model-based risk estimates are likely too narrow, as there remains residual misclassification of long COVID outcomes in that cohort not factored into the confidence interval boundaries. We did not distinguish between vaccine types, though previous studies and initial tabulations failed to detect significant differences in their associations with long COVID[17–19]. The ICD-10 code for long COVID, U09.9, was not implemented until October 2021, and it has not been fully adopted. The previously recommended ICD-10 code, B94.8, is more general and is used to diagnose long-term complications from any viral infection. We accepted B94.8 as a long COVID diagnosis because the use of the code in our data by mid-2021 was 40 times higher than its baseline use in 2018 and 2019. A sensitivity analysis using only U09.9 returned nearly identical results.

In conclusion, vaccination was consistently associated with lower odds of both a long COVID clinical diagnosis as well as a high-confidence computationally derived diagnosis, regardless of viral epoch and taking into account age, sex, and demographics. This multi-method strategy provides additional evidence on the controversial and yet understudied and challenging topic of whether vaccination reduces the risk of long COVID.

## Methods
### Base population
This study is part of the NIH Researching COVID to Enhance Recovery (RECOVER) Initiative, which seeks to understand, treat, and prevent PASC. For more information on RECOVER, visit https://recovercovid.org. All analyses described here were performed within the secure N3C Data Enclave. N3C's methods for patient identification, data acquisition, ingestion, data quality assessment, and harmonization have been described previously in refs. 13,34. The study population was drawn from 5,434,528 COVID-19-positive patients available in N3C. A COVID-19 index date (index) was defined as the earliest recorded indication of COVID-19 infection. Individuals who met the following inclusion criteria were eligible: (1) having an International Classification of Diseases-10-Clinical Modification (ICD-10) COVID-19 diagnosis code (U07.1) or a positive SARS-CoV-2 PCR or antigen test between August 1, 2021 and January 31, 2022; (2) having a recorded health care visit between 120 and 300 days after index; (3) having at least two recorded health care visits in the year prior to index; (4) being ≥18 years old at index; and (5) having either completed or not started a COVID-19 vaccine regimen at index. One exclusion criterion for a clinical cohort is detailed in the outcome definitions. The end of individuals' follow-up periods varied according to when their healthcare providers last submitted new data, ranging from June 10, 2022 to August 1, 2022.

A known limitation of EHR data is that only those healthcare encounters and services provided by the specific health system are available in the data[35]. The proportion of patients with a recorded vaccination at a given healthcare site is driven by two factors: (1) the true rate of vaccination among the population served and (2) how consistently vaccines are captured by the site. Some sites report no vaccinations, while others sync vaccination records with their state's vaccine registry. There is no explicit indicator of non-vaccination in the N3C Data Enclave, but sites with better-recorded vaccine ratios offer more confidence that patients with no recorded vaccine exposure are unvaccinated. We calculated the recorded vaccine ratio at each site as

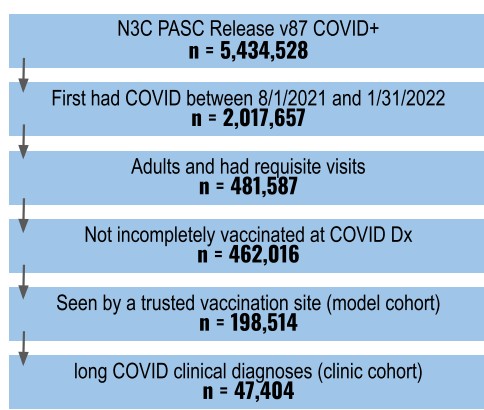

**Fig. 3 | Cohort definition flowchart.** Cumulative number of patients meeting the study's inclusion criteria.

the ratio of two statistics: the observed proportion of patients with a vaccination record and an expected vaccination rate derived from CDC reporting[36] for the population served. Sites with an observed proportion of at least two-thirds of their expected vaccination rate were eligible for analysis, leaving 198,514 patients at eleven sites that met our inclusion criteria. A full breakdown of how many patients met our inclusion criteria is shown in Fig. 3.

As much as possible, we account for confounding due to sex through the inverse probability of treatment weighting and by including sex as a covariate in all primary models. Demographics were defined through standard concepts available in the Observational Medical Outcomes Partnership (OMOP) Common Data Model (CDM)[37,38]. Gender is not available in the CDM[39], and was therefore not considered in the study.

### Exposure definition
Those who completed their vaccine regimen (two doses of BNT162b2 or mRNA-1273 or a single dose of Ad26.COV2.S) two weeks prior to the index were considered vaccinated, while those with no recorded vaccines at the index were considered unvaccinated. Partially vaccinated patients at index failed to meet the fifth inclusion criterion.

### Outcome definitions
Definitions of long COVID vary. The CDC defines post-COVID symptoms as those beginning at least 4 weeks after infection[40], while the WHO defines long COVID as beginning "usually 3 months" from COVID-19 onset with symptoms lasting more than 2 months not explained by another condition[6]. We use two definitions, one each for the clinic and model-based cohorts, that balance these organizational definitions with the strength of evidence available in each cohort.

**Clinical definition.** We considered three clinical indicators of long COVID: (1) an ICD-10 code for post-COVID-19 condition (U09.9), (2) an ICD-10 code for sequelae of other specific infectious and parasitic diseases (B94.8), or (3) a visit to a long COVID clinic. Prior to the introduction of U09.9 in October 2021, the CDC endorsed B94.8 to indicate long-term complications of SARS-CoV-2 infection. As with vaccination, not all sites report clinical indicators of long COVID. Six out of eleven sites, comprising 47,404 of 198,514 eligible patients, submitted clinical indicators of long COVID for at least 250 patients. We used patients from these six sites to form a clinic-based cohort of patients, whom we deemed eligible for receiving a clinical long COVID indicator. The clinic-based cohort has one additional exclusion criteria: those with a long COVID clinical indicator within 45 days of the index were omitted, because diagnoses within this time period are less likely to align with the generally accepted long COVID definitions.

Any long COVID clinical indicator was sufficient to label a patient as having had long COVID in the logistic regression. If patients had multiple encounters with a clinical indicator of long COVID, the earliest was used as the event date for purposes of the time-to-event analysis. Death and COVID-19 vaccination after COVID-19 onset were censoring events.

**Model-based definition.** Long COVID was classified in the model-based cohort as a computational phenotype (CP) using the long COVID cohort identification machine learning model described in ref. 10. A CP is a model trained on EHR data, which can be used to infer the likelihood that a patient has a phenotype (in this case, long COVID) based on their clinical history[41]. For the purposes of this study, the CP model was retrained with U09.9 diagnoses as the target event and without vaccination status as an input. The model calculates a long COVID likelihood score (range 0 to 1) for each patient beginning 100 days after the index using only conditions and drugs observed as of that day. New scores are generated in 30-day intervals until 300 days after the index or June 1, 2022, whichever comes first. Patients scoring above 0.9 in any interval were labeled as having long COVID. A threshold of 0.9 was chosen as it resulted in a similar prevalence of long COVID across the model-based and clinic-based outcomes. The earliest interval receiving a score above 0.9 was assigned as the event date for purposes of the time-to-event analysis. As in the clinic-based definition, death and COVID-19 vaccination were censoring events.

Any patient meeting our inclusion criteria from any of the eleven sites was eligible for a model-derived indicator of long COVID and was included in the model-based cohort. Therefore, all patients in the clinic-based cohort are also included in the model-based cohort, where they can (and sometimes do) have a different assigned long COVID outcome. This is not unexpected—the CP model was trained using U09.9 as the target, while we include U09.9, B94.8, and long COVID clinic visits as valid clinical diagnoses. Both labels are rare and imperfect; we do not expect one indication to guarantee the other.

### Institutional review board oversight
The N3C data transfer to the United States National Center for Advancing Translational Sciences (NCATS) was approved under a Johns Hopkins University Reliance Protocol #IRB00249128 or individual site agreements with NIH. The use of human data for this study was approved by the Johns Hopkins Medicine Institutional Review Board (IRB) #IRB00279988 through a data use agreement entitled "Characterization of long-COVID: definition, stratification, and multimodal analysis". The N3C Data Enclave is managed under the authority of the NIH; information can be found at https://ncats.nih.gov/n3c/resources.

### Statistical analysis
Two analyses were carried out to estimate the association between vaccination and long COVID: (1) logistic regression to calculate an overall association while controlling for patient characteristics and (2) Cox proportional hazards to incorporate differences in the time-to-event for long COVID. We consider both analyses as primary, as each has its own strengths and weaknesses. Proportional hazards uses censoring to account for varying follow-up horizons but require a date for a long COVID diagnosis and for hazard functions to be proportional over time. Logistic regression considers varying follow-up horizons through indicators of acute COVID-19 onset timing, but does not explicitly model times-to-event as done in proportional hazards. We present the results of both analyses as a test of the robustness of the association.

Given our requirement that long COVID diagnoses occur at least 45 days after index, our proportional hazards model uses index + 45 days as the beginning of the modeled time period.

Inverse probability of treatment weighting (IPTW) was applied to both logistic regression and proportional hazards to control for differences in patient characteristics across the vaccinated and

unvaccinated groups. IPTW is a method to adjust for confounding (covariates which affect both the treatment and the outcome) in observational studies. IPTW creates a pseudo-cohort in which the likelihood of treatment is independent of the measured covariates[42]. Logistic regression was used to estimate a treatment propensity score based on sex, demographics, medical history, social determinants of health, and spatial and temporal variables. Our selection of covariates was informed by the literature on important indicators of long COVID[10,43,44]. Covariate balance before and after weighting was evaluated with standardized mean differences. Covariates with a standardized mean difference of less than 0.1 were considered well-balanced. Stabilized treatment weights were calculated as outlined in ref. 45. Standard errors in the IPTW-adjusted models were calculated from 200 bootstrapped iterations based on the standard deviation of the estimates[46]. Unadjusted associations were also calculated and reported.

For logistic regression models, studentized residuals, leverage scores, Cook's distances, and DFBETAS were examined to identify influential observations. Residual analysis helps to identify whether the regression assumption of homogeneous variance is violated. The other statistics identify observations which have an outsized influence on model parameters, which may indicate that the model is unstable[47]. For proportional hazards models, the Lifelines package's *CoxPHFitter.check_assumptions* method was used to test the assumption that each covariate's effect on the hazard rate is constant over time[48,49]. Interactions with time were added to the model for covariates which did not meet the proportional hazards assumption. Variables with more than two levels were binned and represented through indicators. Any indicator with fewer than ten patients identified as having long COVID for a given analysis is not used in that analysis.

### Sensitivity analyses

Six sensitivity analyses were conducted. The first four use the same cohorts as the primary analyses. They test the sensitivity of the IPTW-adjusted and unadjusted vaccination status coefficients in the logistic regression and proportional hazards models across four dimensions: (1) CP score threshold (0.3 to 0.95), (2) with or without independent features in addition to vaccination, (3) including or not including post-index vaccinations as a censoring event, and (4) using only U09.9 diagnoses to label long COVID. The first sensitivity dimension was not relevant for the clinic-based outcome, the third was not relevant for logistic regression analyses, and the fourth was not relevant for the model-based outcome.

The fifth and sixth sensitivity analyses used modified cohorts. The fifth analysis included only patients from partner facilities with the highest recorded vaccine ratios (≥89%). Four sites have recorded vaccine ratios of 89–90%; the next highest is 78%. The sixth sensitivity analysis eliminated the requirement for a recorded healthcare visit after COVID-19 infection and censored individuals after their last visit. Those without a visit after COVID-19 were censored the day after the index. For this analysis, the modeled time period began the day after the index (instead of 45 days after the index in the primary analysis), though individuals remained ineligible for long COVID designation until 45 days after the index. The sixth analysis was only relevant for the proportional hazards model in the clinic-based analysis, as the computable phenotype model requires a post-COVID-19 visit at least 60 days after COVID-19 and censoring is not available in logistic regression models[10]. Vaccination propensity scores, as well as the model coefficients, were recalculated for each of the modified cohorts in the fifth and sixth sensitivity analyses.

### Subanalysis

A subanalysis was performed to determine if the time of vaccination relative to acute COVID-19 diagnosis severely modulates the association between vaccination and long COVID diagnosis. Both primary analyses were repeated with the addition of indicators for the number of weeks between an individual's last pre-COVID-19 vaccination and their COVID-19 diagnosis date.

All analyses were conducted using Python (version 3.6.10) with the Statsmodels (0.12.2) and Lifelines (0.26.4) packages. Preprocessing was done in R (3.5.1) and Python (3.6.10) with the PySpark (3.2.1), pandas (0.25.3), and numpy (1.19.5) packages. Study design elements, methods, and results were reported as consistent with STROBE guidelines[50].

### Reporting summary

Further information on research design is available in the Nature Portfolio Reporting Summary linked to this article.

## Data availability

All data is available in the N3C Data Enclave to those with an approved protocol and data use request from an institutional review board. Data access is governed under the authority of the National Institutes of Health; more information on accessing the data can be found at https://covid.cd2h.org/for-researchers. See Haendel et al. for additional detail on how data is ingested, managed, and protected within the N3C Data Enclave[13].

## Code availability

The N3C Enclave is available for public use. To access data used within this manuscript, institutions must have a signed Data Use Agreement executed with the US National Center for Advancing Translational Sciences (NCATS) and their investigators must complete mandatory training and must submit a Data Use Request (DUR) to N3C. To request N3C data access, researchers must follow the instructions at https://covid.cd2h.org/onboarding. Code is available to those with valid login credentials for the N3C Data Enclave. It was written for use in the enclave on the Palantir Foundry platform[51], where the analysis can be reproduced by researchers. It can be exported for review upon request.

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

## Acknowledgements

Authorship was determined using ICMJE recommendations. The content is solely the responsibility of the authors and does not necessarily represent the official views of the National Institutes of Health or N3C or RECOVER. This study is part of the National Institutes of Health (NIH) Researching COVID to Enhance Recovery (RECOVER) Initiative, which seeks to understand, treat, and prevent the post-acute sequelae of SARS-CoV-2 infection (PASC). For more information on RECOVER, visit https://recovercovid.org/. This research was supported by the NIH Agreement OTA OT2HL161847 as part of the RECOVER research

program (to all authors) as well as Axle Informatics Subcontract NCATS-P00438-B (to A.G.Z.) and iTHRIV Integrated Translational Health Research Institute of Virginia UL1TR003015 (to A.G.Z.). We would like to thank the National Community Engagement Group (NCEG), all patients, caregivers, and community Representatives, and all the participants enrolled in the RECOVER Initiative. N3C Attribution: The analyses described in this publication were conducted with data or tools accessed through the NCATS N3C Data Enclave covid.cd2h.org/enclave and supported by CD2H - The National COVID Cohort Collaborative (N3C) IDeA CTR Collaboration 3U24TR002306-04S2 NCATS U24 TR002306. This research was possible because of the patients whose information is included within the data from participating organizations (covid.cd2h.org/dtas) and the organizations and scientists (covid.cd2h.org/duas) who have contributed to the on-going development of this community resource (cite this https://doi.org/10.1093/jamia/ocaa196). We gratefully acknowledge the following core-contributors to N3C: Adam B. Wilcox, Adam M. Lee, Alexis Graves, Alfred (Jerrod) Anzalone, Amin Manna, Amit Saha, Amy Olex, Andrea Zhou, Andrew E. Williams, Andrew Southerland, Andrew T. Girvin, Anita Walden, Anjali A. Sharathkumar, Benjamin Amor, Benjamin Bates, Brian Hendricks, Brijesh Patel, Caleb Alexander, Carolyn Bramante, Cavin Ward-Caviness, Charisse Madlock-Brown, Christine Suver, Christopher Chute, Christopher Dillon, Chunlei Wu, Clare Schmitt, Cliff Takemoto, Dan Housman, Davera Gabriel, David A. Eichmann, Diego Mazzotti, Don Brown, Eilis Boudreau, Elaine Hill, Elizabeth Zampino, Emily Carlson Marti, Emily R. Pfaff, Evan French, Farrukh M Koraishy, Federico Mariona, Fred Prior, George Sokos, Greg Martin, Harold Lehmann, Heidi Spratt, Hemalkumar Mehta, Hongfang Liu, Hythem Sidky, J.W. Awori Hayanga, Jami Pincavitch, Jaylyn Clark, Jeremy Richard Harper, Jessica Islam, Jin Ge, Joel Gagnier, Joel H. Saltz, Joel Saltz, Johanna Loomba, John Buse, Jomol Mathew, Joni L. Rutter, Julie A. McMurry, Justin Guinney, Justin Starren, Karen Crowley, Katie Rebecca Bradwell, Kellie M. Walters, Ken Wilkins, Kenneth R. Gersing, Kenrick Dwain Cato, Kimberly Murray, Kristin Kostka, Lavance Northington, Lee Allan Pyles, Leonie Misquitta, Lesley Cottrell, Lili Portilla, Mariam Deacy, Mark M. Bissell, Marshall Clark, Mary Emmett, Mary Morrison Saltz, Matvey B. Palchuk, Melissa A. Haendel, Meredith Adams, Meredith Temple-O'Connor, Michael G. Kurilla, Michele Morris, Nabeel Qureshi, Nasia Safdar, Nicole Garbarini, Noha Sharafeldin, Ofer Sadan, Patricia A. Francis, Penny Wung Burgoon, Peter Robinson, Philip R.O. Payne, Rafael Fuentes, Randeep Jawa, Rebecca Erwin-Cohen, Rena Patel, Richard A. Moffitt, Richard L. Zhu, Rishi Kamaleswaran, Robert Hurley, Robert T. Miller, Saiju Pyarajan, Sam G. Michael, Samuel Bozzette, Sandeep Mallipattu, Satyanarayana Vedula, Scott Chapman, Shawn T. O'Neil, Soko Setoguchi, Stephanie S. Hong, Steve Johnson, Tellen D. Bennett, Tiffany Callahan, Umit Topaloglu, Usman Sheikh, Valery Gordon, Vignesh Subbian, Warren A. Kibbe, Wenndy Hernandez, Will Beasley, Will Cooper, William Hillegass, and Xiaohan Tanner Zhang. Details of contributions available at covid.cd2h.org/core-contributors. Data Partners: The following institutions whose data is released or pending: Available: Advocate Health Care Network—UL1TR002389: The Institute for Translational Medicine (ITM) • Boston University Medical Campus—UL1TR001430: Boston University Clinical and Translational Science Institute • Brown University—U54GM115677: Advance Clinical Translational Research (Advance-CTR) • Carilion Clinic—UL1TR003015: iTHRIV Integrated Translational health Research Institute of Virginia • Charleston Area Medical Center—U54GM104942: West Virginia Clinical and Translational Science Institute (WVCTSI) • Children's Hospital Colorado—UL1TR002535: Colorado Clinical and Translational Sciences Institute • Columbia University Irving Medical Center—UL1TR001873: Irving Institute for Clinical and Translational Research • Duke University—UL1TR002553: Duke Clinical and Translational Science Institute • George Washington Children's Research Institute— UL1TR001876: Clinical and Translational Science Institute at Children's National (CTSA-CN) • George Washington University—

UL1TR001876: Clinical and Translational Science Institute at Children's National (CTSA-CN) • Indiana University School of Medicine—UL1TR002529: Indiana Clinical and Translational Science Institute • Johns Hopkins University—UL1TR003098: Johns Hopkins Institute for Clinical and Translational Research • Loyola Medicine—Loyola University Medical Center • Loyola University Medical Center—UL1TR002389: The Institute for Translational Medicine (ITM) • Maine Medical Center—U54GM115516: Northern New England Clinical & Translational Research (NNE-CTR) Network • Massachusetts General Brigham—UL1TR002541: Harvard Catalyst • Mayo Clinic Rochester—UL1TR002377: Mayo Clinic Center for Clinical and Translational Science (CCaTS) • Medical University of South Carolina—UL1TR001450: South Carolina Clinical & Translational Research Institute (SCTR) • Montefiore Medical Center—UL1TR002556: Institute for Clinical and Translational Research at Einstein and Montefiore • Nemours—U54GM104941: Delaware CTR ACCEL Program • NorthShore University HealthSystem—UL1TR002389: The Institute for Translational Medicine (ITM) • Northwestern University at Chicago—UL1TR001422: Northwestern University Clinical and Translational Science Institute (NUCATS) • OCHIN—INV-018455: Bill and Melinda Gates Foundation grant to Sage Bionetworks • Oregon Health & Science University—UL1TR002369: Oregon Clinical and Translational Research Institute • Penn State Health Milton S. Hershey Medical Center—UL1TR002014: Penn State Clinical and Translational Science Institute • Rush University Medical Center—UL1TR002389: The Institute for Translational Medicine (ITM) • Rutgers, The State University of New Jersey—UL1TR003017: New Jersey Alliance for Clinical and Translational Science • Stony Brook University— U24TR002306 • The Ohio State University—UL1TR002733: Center for Clinical and Translational Science • The State University of New York at Buffalo—UL1TR001412: Clinical and Translational Science Institute • The University of Chicago—UL1TR002389: The Institute for Translational Medicine (ITM) • The University of Iowa—UL1TR002537: Institute for Clinical and Translational Science • The University of Miami Leonard M. Miller School of Medicine—UL1TR002736: University of Miami Clinical and Translational Science Institute • The University of Michigan at Ann Arbor—UL1TR002240: Michigan Institute for Clinical and Health Research • The University of Texas Health Science Center at Houston— UL1TR003167: Center for Clinical and Translational Sciences (CCTS) • The University of Texas Medical Branch at Galveston—UL1TR001439: The Institute for Translational Sciences • The University of Utah—UL1TR002538: Uhealth Center for Clinical and Translational Science • Tufts Medical Center—UL1TR002544: Tufts Clinical and Translational Science Institute • Tulane University—UL1TR003096: Center for Clinical and Translational Science • University Medical Center New Orleans—U54GM104940: Louisiana Clinical and Translational Science (LA CaTS) Center • University of Alabama at Birmingham—UL1TR003096: Center for Clinical and Translational Science • University of Arkansas for Medical Sciences—UL1TR003107: UAMS Translational Research Institute • University of Cincinnati—UL1TR001425: Center for Clinical and Translational Science and Training • University of Colorado Denver, Anschutz Medical Campus— UL1TR002535: Colorado Clinical and Translational Sciences Institute • University of Illinois at Chicago—UL1TR002003: UIC Center for Clinical and Translational Science • University of Kansas Medical Center—UL1TR002366: Frontiers: University of Kansas Clinical and Translational Science Institute • University of Kentucky—UL1TR001998: UK Center for Clinical and Translational Science • University of Massachusetts Medical School Worcester—UL1TR001453: The UMass Center for Clinical and Translational Science (UMCCTS) • University of Minnesota—UL1TR002494: Clinical and Translational Science Institute • University of Mississippi Medical Center—U54GM115428: Mississippi Center for Clinical and Translational Research (CCTR) • University of Nebraska Medical Center—U54GM115458: Great Plains IDeA-Clinical & Translational Research • University of North Carolina at Chapel Hill—UL1TR002489: North Carolina Translational and Clinical Science

Institute • University of Oklahoma Health Sciences Center—U54GM104938: Oklahoma Clinical and Translational Science Institute (OCTSI) • University of Rochester—UL1TR002001: UR Clinical & Translational Science Institute • University of Southern California—UL1TR001855: The Southern California Clinical and Translational Science Institute (SC CTSI) • University of Vermont—U54GM115516: Northern New England Clinical & Translational Research (NNE-CTR) Network • University of Virginia—UL1TR003015: iTHRIV Integrated Translational health Research Institute of Virginia • University of Washington—UL1TR002319: Institute of Translational Health Sciences • University of Wisconsin-Madison—UL1TR002373: UW Institute for Clinical and Translational Research • Vanderbilt University Medical Center— UL1TR002243: Vanderbilt Institute for Clinical and Translational Research • Virginia Commonwealth University— UL1TR002649: C. Kenneth and Dianne Wright Center for Clinical and Translational Research • Wake Forest University Health Sciences—UL1TR001420: Wake Forest Clinical and Translational Science Institute • Washington University in St. Louis—UL1TR002345: Institute of Clinical and Translational Sciences • Weill Medical College of Cornell University—UL1TR002384: Weill Cornell Medicine Clinical and Translational Science Center • West Virginia University—U54GM104942: West Virginia Clinical and Translational Science Institute (WVCTSI). Submitted: Icahn School of Medicine at Mount Sinai—UL1TR001433: ConduITS Institute for Translational Sciences • The University of Texas Health Science Center at Tyler—UL1TR003167: Center for Clinical and Translational Sciences (CCTS) • University of California, Davis— UL1TR001860: UC Davis Health Clinical and Translational Science Center • University of California, Irvine— UL1TR001414: The UC Irvine Institute for Clinical and Translational Science (ICTS) • University of California, Los Angeles—UL1TR001881: UCLA Clinical Translational Science Institute • University of California, San Diego—UL1TR001442: Altman Clinical and Translational Research Institute • University of California, San Francisco—UL1TR001872: UCSF Clinical and Translational Science Institute. Pending: Arkansas Children's Hospital—UL1TR003107: UAMS Translational Research Institute • Baylor College of Medicine—None (Voluntary) • Children's Hospital of Philadelphia—UL1TR001878: Institute for Translational Medicine and Therapeutics • Cincinnati Children's Hospital Medical Center—UL1TR001425: Center for Clinical and Translational Science and Training • Emory University—UL1TR002378: Georgia Clinical and Translational Science Alliance • HonorHealth—None (Voluntary) • Loyola University Chicago—UL1TR002389: The Institute for Translational Medicine (ITM) • Medical College of Wisconsin—UL1TR001436: Clinical and Translational Science Institute of Southeast Wisconsin • MedStar Health Research Institute—UL1TR001409: The Georgetown-Howard Universities Center for Clinical and Translational Science (GHUCCTS) • MetroHealth—None (Voluntary) • Montana State University—U54GM115371: American Indian/Alaska Native CTR • NYU Langone Medical Center—UL1TR001445: Langone Health's Clinical and Translational Science Institute • Ochsner Medical Center—U54GM104940: Louisiana Clinical and Translational Science (LA CaTS) Center • Regenstrief Institute—UL1TR002529: Indiana Clinical and Translational Science Institute • Sanford Research—None (Voluntary) • Stanford University—UL1TR003142: Spectrum: The Stanford Center for Clinical and Translational Research and Education • The Rockefeller University—UL1TR001866: Center for Clinical and Translational Science • The Scripps Research Institute—UL1TR002550: Scripps Research Translational Institute • University of Florida—UL1TR001427: UF Clinical and Translational Science Institute • University of New Mexico Health Sciences Center—UL1TR001449: University of New Mexico Clinical and Translational Science Center • University of Texas Health Science Center at San Antonio—UL1TR002645: Institute for Integration of Medicine and Science • Yale New Haven Hospital—UL1TR001863: Yale Center for Clinical Investigation

## Author contributions

M.D.B., R.F.C., A.J.P., and E.C.H. performed the data analysis. S.R. provided substantive input to the interpretation and conclusions as a patient representative. M.D.B., R.F.C., A.J.P., E.C.H., S.R., J.A.M., E.R.P., M.A.H., and C.G.C. wrote and revised the manuscript. M.D.B., R.F.C., P.J.L., A.T.G., M.C., A.G.Z., R.A.M., and E.R.P. contributed significantly to cohort definition and creation. M.J.F. advised in the design of the statistical analysis and revised the manuscript. M.A.H. and C.G.C. supervised the study.

## Competing interests

The authors declare no competing interests.

## Additional information

## N3C

Til Stürmer[4], Johanna J. Loomba[7] & Farrukh M. Koraishy[11]

[11]Stony Brook Medicine, Stony Brook, NY, USA.

## RECOVER Consortia

Jasmin Divers[12], Lorna E. Thorpe[13], Leora Horwitz[13] & Stuart Katz[14]

[12]New York University Long Island School of Medicine, Department of Foundations of Medicine, Mineola, NY, USA. [13]New York University Grossman School of Medicine, Department of Population Health, New York, NY, USA. [14]New York University Grossman School of Medicine, Leon H. Charney Division of Cardiology, Department of Medicine, New York, NY, USA.

