## [Peer Review File · Nature Communications]

Long COVID risk and pre-COVID vaccination in an EHR-based cohort study from the RECOVER programREVIEWER COMMENTS

Reviewer #1 (Remarks to the Author):

This study from the RECOVER initiative examines the association of vaccination with long COVID. While previous studies have already examined this, it is important to examine from different avenues of approach to bolster evidence in support of informed decision making. Overall I felt like, within the studies aims, the analytic approach was appropriate. I appreciated the wide discussion of limitations.

Have just a couple comments, I consider minor, that I hope are helpful:

- 1) One of the more unique aspects of this study compared to prior studies was the focus on examining the long COVID outcome via a clinical diagnosis. It might be helpful for readers to provide some further explanation as to why this is important vis-a-vis defining based on symptomology alone; why was this approach taken as compared to focusing on symptomology given the tradeoffs?
- 2) If there are systematic differences (bias) in date of diagnoses, logistic regressions could be affected by this too, as it would change the event rate. As a summary measure, just like an HR not modeled as time varying, it would also be impacted by the horizon of follow-up if the PH assumption was violated. So I am not sure that these two analyses necessarily address each others limitations; they simply provide a different approach with how timing is handled.
- 3) It might be worth considering for clarity referring to it as "model based outcome" rather than "model based cohort."

Reviewer #2 (Remarks to the Author):

This paper examines the association between vaccination with a COVID19 vaccine prior to infection (exposure) and the presence of Long COVID in patients previously infected with SARS CoV-2 (outcome). While there is an emerging consensus in the literature that there is indeed a protective effect of vaccination, this study adds to the existing evidence for two main reasons a) the size of the study sample (100,000 + vs a few hundreds/thousands in most existing studies) and b) the fact the outcome is extracted from medical records as opposed to self reported symptoms. The study finds that pre-infection vaccination is associated with a decreased incidence of Long COVID, consistently with the literature. The statistical analysis is sound, if complex (see comments below). There are however a few concerns that need to be addressed before considering whether the paper can be published. The most serious issue are around misclassification of both exposure and outcome.

-The paper is very stats heavy- it could be a reflection of the skillset of the research team, perhaps more focused on data science than epidemiology. As a result, the most likely readership (infectious disease epidemiologist, public health planners) may not be familiar with some of the statistical tools that are mentioned but not explained. This includes computational phenotypes, DFBETA, Cook's distance, IPTW. These tools must be spelled out and explained in plain language

-I have concerns about the quality and validity of the dataset. It is concerning that from a dataset of 5.5 million records, less than 50,000 (<1%) meet the criteria for inclusion. Figure 1 shows the process, and while some criteria for exclusion are obviously valid (time period for example), my main concern is the completeness and accuracy of vaccination status- the main exposure, as described lines 217-227. The inclusion of sites is based on an aggregate cutoff of at least two thirds of the expected vaccine coverage, leading to the inclusion of just under 200,000 records out of over 2 million in the available time period. More over the threshold for inclusion is very low- two thirds correct means that up to 33% of records included in the study have an incorrectly recorded vaccinated status. The authors use a quality indicator on aggregated data and apply it to the individual health record, which is the unit of analysis- an example of ecological fallacy. The authors must explain more clearly how they dealt with this level of uncertainty and explain how the results can be full trusted despite this data limitation

-There is also an issue with classification according to outcome- as the authors note, the

prevalence in their cohort (<2%) is much lower than what is reported in the literature (10-30%)- primarily, as the authors mention, because they rely on the clinical records meaning only patients who sought medical help for their symptoms can be classified as long COVID- most likely not the majority of patients. This implies that any patients who is experiencing Long COVID symptoms according to the WHO case definition but did not seek medical help will be classified as not having Long COVID- if we apply an average 20% of SARS CoV2 infected patients who develop long COVID, this would imply that approximately 35,000 individuals would be misclassified according to outcome- likely biasing the association towards the null. The authors need to explain whether this issue affects their conclusion

-The background is very limited and lacks key information about long COVID- its prevalence, key features, and more importantly whether the inclusion of long COVID ICD code and CP implies the use of a particular case definition. There is no mention of the most used case definition, the WHO one (see A clinical case definition of post-COVID-19 condition by a Delphi consensus - PubMed (nih.gov))

One of the key features of this case definition is that it differentiates between long COVID and prolonged acute episode. One of the key features of this case definition is that symptoms usually occur 3 months from the onset, with symptoms that last for at least 2 months and cannot be explained by an alternative diagnosis. However in figure 2 it seems that the authors have considered any ICD code or CP identified case from the day of disease onset, not taking in consideration the lag between acute symptoms and PASC, and not taking symptom duration into account. This could lead to overestimating the incidence of Long COVID and conflating vaccine effectiveness against acute infection and long COVID. The authors may want to consider excluding PASC diagnosis within the first 30 or 60 days of infection. I am not sure if the dataset enables to estimate the duration of symptoms, but if it can it would add tremendous weight to the paper to focus on cases that fit the prevalent case definition

-From the vaccine effectiveness data against infection and acute disease severity, we know that protection wanes rapidly over time- therefore the time elapsed between vaccination and disease may be a highly relevant factor. Did the authors consider taking this variable into consideration? Equally individuals vaccinated immediately prior to infection (10-14 days before?) should not be classified as vaccination

-Did the authors conduct a sub analysis considering separately those who received 2 doses vs 3 doses+? I am not sure why the authors consider one viral vector vaccine to be a completed course- while this may be the original recommendation for the J&J vaccine, this is not the case for the AZ vaccine. Patients who received one dose of an AZ vaccine should not be considered fully vaccinated.

Reviewer #3 (Remarks to the Author):

Review

In this paper, the authors seek to estimate the association between pre-COVID vaccination and long COVID incidence. As the authors mentioned in their introduction, this question has been addressed elsewhere (e.g. in Al Aly et al 2022 and Taquet et al 2022). However, the question is far from answered in any definitive way and the present study has one unique feature: it is based on sites wherein recorded vaccination rates is close (> 66%) to what is expected.

It also has some important limitations which can hopefully be addressed as sensitivity analyses or further discussion.

1) The time of the index event (i.e. the month of the COVID-19 infection) is an important covariate and while it is "adjusted for" within the IPTW, it would be important to also run an analysis stratified by month of index event. This is because the time of the index event does not just affect the propensity of exposure, but also the duration of follow-up (and hence the probability of being censored).

2) The lack of adjustment for psychiatric comorbidities/history thereof is puzzling, given that (a) psychiatric symptoms are often considered to be part of long-COVID, and (b) psychiatric comorbidities affect patients' healthcare utilization. This should be incorporated in the analysis.

3) The inclusion of "visit to a long COVID clinic" in the primary outcome (clinical definition) affects the validity of that outcome as patients might attend long COVID clinic only to be told they do not have long COVID. It is unclear which individual outcome (U09.9, B94.8, or visit to an LC clinic) drive the incidence and hazard ratios for the primary composite outcome. Could the authors (a) report the total number of each of the three clinical events observed in their cohorts, and (b) report the results of a sensitivity analysis in which each event of the composite outcome is assessed individually?

4) The actual incidence of long COVID is unknown. But of all incidences reported, the one reported here (1.5%) is among the lowest. This is not surprising given that many clinicians are probably not yet using the U09.9 code. One could therefore consider that the computational phenotypes with lower thresholds might better capture the proportion of patients with long COVID. The fact that little effect of vaccination is seen when the CP threshold is decreased calls for more cautious conclusion in the abstract (hence the need to rephrase the last sentence).

5) In the primary analysis, patients had to have had a healthcare visit between 120 and 300 days after index. This implies that the results are not valid for the patients not making one such visit (which, according to Fig 1 might constitute the majority of patients). Could the authors run a sensitivity analysis in which all patients are included and in which they can be censored at their last visit (even if this occurs before Day 120) or censored at day 300 if their last visit was made over 300 days after the index?

6) Death was a censoring condition in the primary analysis. This raises the possibility of survivorship bias as vaccinated individuals are less likely to die from COVID. Could the authors run a sensitivity analysis in which death is included in a composite endpoint with long COVID to assess its influence on the primary results?

7) One limitation of the use of a clinical diagnosis or computational phenotype is that it does not discern between the different symptoms of long-COVID. Both studies by Al Aly et al, and Taquet et al showed that the 'protective effect' of prior vaccination varied between symptoms. This should be better highlighted in the discussion.

Response to Referees

Reviewer #1

This study from the RECOVER initiative examines the association of vaccination with long COVID. While previous studies have already examined this, it is important to examine from different avenues of approach to bolster evidence in support of informed decision making.

Overall I felt like, within the studies aims, the analytic approach was appropriate. I appreciated the wide discussion of limitations.

Have just a couple comments, I consider minor, that I hope are helpful:

Comment #1: One of the more unique aspects of this study compared to prior studies was the focus on examining the long COVID outcome via a clinical diagnosis. It might be helpful for readers to provide some further explanation as to why this is important vis-a-vis defining based on symptomology alone; why was this approach taken as compared to focusing on symptomology given the tradeoffs?

Response #1: We agree, and have added the following text to the Discussion:

“Our use of long COVID diagnosis and a computable phenotype as outcomes differentiate this study from others (Al-Aly, Tarquet, Simon), which measure the association between vaccination and a curated list of long COVID symptoms. Each approach has its strengths. Our clinical outcomes reduce measurement error due to false positives (e.g., long COVID symptoms caused by something other than long COVID). However, other studies show that long COVID symptoms differ in their relationship with vaccination. Our outcomes obscure such variation. We conclude that it is beneficial to study this relationship from both perspectives.”

Comment #2: If there are systematic differences (bias) in date of diagnoses, logistic regressions could be affected by this too, as it would change the event rate. As a summary measure, just like an HR not modeled as time varying, it would also be impacted by the horizon of follow-up if the PH assumption was violated. So I am not sure that these two analyses necessarily address each others limitations; they simple provide a different approach with how timing is handled.

Response #2: This is a good clarifying suggestion. We have rewritten this section of the Methods to no longer state that the two techniques address one another’s limitations, but instead that they offer two different but valuable approaches:

“We consider both analyses as primary, as each has its own strengths and weaknesses. Proportional hazards uses censoring to account for varying follow-up horizons but requires a date for long COVID

diagnosis and for hazard functions to be proportional over time. Logistic regression considers varying follow-up horizons through indicators of acute COVID-19 onset timing, but does not explicitly model times-to-event as done in proportional hazards. We present the results of both analyses as a test of the robustness of the association.”

Comment #3: It might be worth considering for clarity referring to it as “model based outcome” rather than “model based cohort.”

Response #3: Thank you for the suggestion. We agree that it is the outcome that is model-based vs clinic-based, but when referring to the cohorts we find it more direct (and still understandable) to refer to them as model-based rather than as using a model-based outcome. However, we have amended our language in several places, so that model-based cohort is used to refer to the cohort, but model-based analysis/results/outcome is used when referring to the analysis/results/outcome.

Reviewer #2

This paper examines the association between vaccination with a COVID19 vaccine prior to infection (exposure) and the presence of Long COVID in patients previously infected with SARS CoV-2 (outcome). While there is an emerging consensus in the literature that there is indeed a protective effect of vaccination, this study adds to the existing evidence for two main reasons a) the size of the study sample (100,000 + vs a few hundreds/thousands in most existing studies) and b) the fact the outcome is extracted from medical records as opposed to self reported symptoms. The study finds that pre-infection vaccination is associated with a decreased incidence of Long COVID, consistently with the literature. The statistical analysis is sound, if complex (see comments below). There are however a few concerns that need to be addressed before considering whether the paper can be published. The most serious issue are around misclassification of both exposure and outcome.

Comment #1: The paper is very stats heavy- it could be a reflection of the skillset of the research team, perhaps more focused on data science than epidemiology. As a result, the most likely readership (infectious disease epidemiologist, public health planners) may not be familiar with some of the statistical tools that are mentioned but not explained. This includes computational phenotypes, DFBETA, Cook's distance, IPTW. These tools must be spelled out and explained in plain language

Response #1: We thank the reviewers for this helpful suggestion. We agree that defining these terms will help make the paper more accessible to the target readership. We have defined these terms as follows in the Methods section.

“Long COVID was classified in the model-based cohort as a computational phenotype (CP) using the long COVID cohort identification machine learning model described in Pfaff et al, 2022. A CP is a model trained on EHR data, which can be used to infer the likelihood that a patient has a phenotype (in this case, long COVID) based on their clinical history (Kirby).”

“For logistic regression models, studentized residuals, leverage scores, Cook's distances, and DFBETAS were examined to identify influential observations. Residual analysis helps to identify whether the regression assumption of homogeneous variance is violated. The other statistics identify observations which have outsized influence on model parameters, which may indicate that the model is unstable (Van Der Meer).”

“Inverse probability of treatment weighting (IPTW) was applied to both logistic regression and proportional hazards to control for differences in patient characteristics across the vaccinated and unvaccinated groups. IPTW is a method to adjust for confounding (covariates which affect both the treatment and the outcome) in observational studies. IPTW creates a pseudo-cohort in which the likelihood of treatment is independent of the measured covariates (Cole).”

Comment #2: I have concerns about the quality and validity of the dataset. It is concerning that from a dataset of 5.5 million records, less than 50,000 (<1%) meet the criteria for inclusion. Figure 1 shows the process, and while some criteria for exclusion are obviously valid (time period for example), my main concern is the completeness and accuracy of vaccination status- the main exposure, as described lines 217-227. The inclusion of sites is based on an aggregate cutoff of at least two thirds of the expected vaccine coverage, leading to the inclusion of just under 200,000 records out of over 2 million in the available time period. More over the threshold for inclusion is very low- two thirds correct means that up to 33% of records included in the study have an incorrectly recorded vaccinated status. The authors use a quality indicator on aggregated data and apply it to the individual health record, which is the unit of analysis- an example of ecological fallacy. The authors must explain more clearly how they dealt with this level of uncertainty and explain how the results can be fully trusted despite this data limitation.

Response #2: Thank you for this suggestion, we agree this is an important limitation of the study. Nondifferential treatment misclassification tends to bias toward the null, while differential misclassification can produce bias in either direction. To assess the effect of treatment misclassification in our study, we have conducted an additional sensitivity analysis using the few partner facilities with very good vaccination data. We believe that the addition of this sensitivity analysis makes our paper stronger, and we thank the reviewer for raising their concern. Please see the following text that has been added to the Methods:

“The fourth [sensitivity] analysis included only patients from partner facilities with the highest recorded vaccine ratios ($\geq 89\%$). Four sites have recorded vaccine ratios of 89–90%; the next highest is 78%.”

And Results:

“Including only sites with the most complete vaccine reporting (with recorded vaccine ratios of at least 89%) resulted in associations similar to or stronger than the four primary associations.”

In the Discussion we interpret this sensitivity analysis and address the concern more broadly:

“Our cohort is further refined by our requirement that the partner facility have reasonably high recorded vaccine ratios, as defined in our Methods. Most facilities fail to achieve recorded vaccine ratios greater than 66%, as they are not the primary provider of vaccinations in their community, do not link to their state’s vaccine registry, or do not consistently record the vaccinations they provide in the EHR. We do not use a facility’s vaccination rate as an individual characteristic in our models, but rather as a facility-wide inclusion criteria. By limiting to partner facilities with a high vaccination rate, as with our utilization criteria, we refine our cohort to be smaller but more data-rich.

We strictly define our study cohort to minimize the underreporting of vaccination and long COVID, though we acknowledge that it is not entirely resolved. Our sensitivity analysis using only sites with the highest recorded vaccine ratios ($\geq 89\%$) offers some evidence that incomplete vaccine records result in a conservative estimate. The cohorts are small (the model-based cohort has 10,122 patients; the clinic-

based cohort has 5,545), resulting in wide confidence intervals that include the primary estimates for every model (eTable 6). However, the mean estimated associations are stronger than our primary results in three of the four models and remain significant in all four models with 95% confidence. We conclude that our primary estimates are likely conservative, but our primary result—that pre-COVID-19 vaccination is associated with a reduced risk of long COVID diagnosis—is not threatened.”

Comment #3: There is also an issue with classification according to outcome- as the authors note, the prevalence in their cohort (<2%) is much lower than what is reported in the literature (10-30%)- primarily, as the authors mention, because they rely on the clinical records meaning only patients who sought medical help for their symptoms can be classified as long COVID- most likely not the majority of patients. This implies that any patients who is experiencing Long COVID symptoms according to the WHO case definition but did not seek medical help will be classified as not having Long COVID- if we apply an average 20% of SARS CoV2 infected patients who develop long COVID, this would imply that approximately 35,000 individuals would be misclassified according to outcome- likely biasing the association towards the null. The authors need to explain whether this issue affects their conclusion

Response #3: This is a major limitation of any study using diagnoses from EHRs. We attempt to limit this through our utilization inclusion criteria. We have added to the Discussion to address this concern:

“We are forced to assume that those without a recorded condition or symptom do not exhibit it, including our exposure (vaccination), our outcome (long COVID diagnosis), and potentially unrecorded reinfections of COVID-19. We take two steps to mitigate the risk of unrecorded records relevant to the outcome: (1) we require that all participants in the study had established care at the partner facility prior to COVID-19 infection, as evidenced by two healthcare visits in the year prior, and (2) we require that all participants in the study were seen at the partner facility at least 120 days after COVID-19 infection. This results in a significantly smaller cohort, and biases the sample towards high utilizers and those with hospitalizations. However, it remains sufficiently large for analysis and has a lower risk that long COVID will go undiagnosed, as patients were active users of the partner facility both before and four months after COVID-19 infection. In the clinic-based cohort, there is an additional requirement that the facilities have a track record of diagnosing long COVID (though variation between doctors remains).”

Comment #4: The background is very limited and lacks key information about long COVID- its prevalence, key features, and more importantly whether the inclusion of long COVID ICD code and CP implies the use of a particular case definition. There is no mention of the most used case definition , the WHO one (see A clinical case definition of post-COVID-19 condition by a Delphi consensus - PubMed (nih.gov))

Response #4: We have added some background information and citations, both to our own and to other’s work regarding Long Covid phenotypic features for readability. We also describe the pursuit of a gold standard definition of Long Covid, both from an EHR/RWD perspective, as well as the WHO and our own conclusions that a gold standard definition of Long Covid requires both patient and clinician reported features. The EHR does not typically include patient reported features, and here our goal is to

use the EHR for surveillance, patient recruitment, and research purposes. Thus, we have leveraged two definitions of Long Covid, a clinical attestation as a diagnostic code as well as the ML-based definition to elucidate the outcomes of vaccination on Long Covid.

Comment #5: One of the key features of this case definition is that it differentiates between long COVID and prolonged acute episode. One of the key features of this case definition is that symptoms usually occur 3 months from the onset, with symptoms that last for at least 2 months and cannot be explained by an alternative diagnosis. However in figure 2 it seems that the authors have considered any ICD code or CP identified case from the day of disease onset, not taking in consideration the lag between acute symptoms and PASC, and not taking symptom duration into account. This could lead to overestimating the incidence of Long COVID and conflating vaccine effectiveness against acute infection and long COVID. The authors may want to consider excluding PASC diagnosis within the first 30 or 60 days of infection. I am not sure if the dataset enables to estimate the duration of symptoms, but if it can it would add tremendous weight to the paper to focus on cases that fit the prevalent case definition.

Response #5: We are particularly grateful for this comment, which highlighted two mistakes we made in writing up our methods that have now been addressed:

1. Although it was already in place, we failed to address that our exposure definition in the clinic-based cohort required the clinical diagnosis of long COVID to occur no earlier than 45 days after COVID-19 diagnosis. In the model-based cohort, the computable phenotype is only computed starting 100 days after COVID-19 diagnosis, at which point they become eligible to receive a long COVID designation. The following text was added to our Outcome Definitions:

“Definitions of long COVID vary. The CDC defines post-COVID symptoms as those beginning at least four weeks after infection (CDC) while the WHO defines long COVID as beginning “usually 3 months” from COVID-19 onset with symptoms lasting more than 2 months not explained by another condition (Soriano). We use two definitions, one each for the clinic and model-based cohorts, that balance these organizational definitions with the strength of evidence available in each cohort.”

“The clinic-based cohort has one additional exclusion criteria: those with a long COVID clinical indicator within 45 days of index were omitted, because diagnoses within this time period are less likely to align with the generally accepted long COVID definitions.”

2. Our survival analysis begins 45 days after COVID index, when people first become eligible for a long COVID diagnosis in the clinic-based cohort. The x-axes of our Kaplan Meier curves in Figure 2 were therefore not correct, as they actually begin 45 days after index rather than 0 days after index (the curves should be shifted 45 days to the right). This has been resolved. The following text has also been added to the Statistical Analysis methods:

“Given our requirement that long COVID diagnoses occur at least 45 days after index, our proportional hazards model uses index + 45 days as the beginning of the modeled time period.”

Comment #6: From the vaccine effectiveness data against infection and acute disease severity, we know that protection wanes rapidly over time- therefore the time elapsed between vaccination and disease may be a highly relevant factor. Did the authors consider taking this variable into consideration? Equally individuals vaccinated immediately prior to infection (10-14 days before?) should not be classified as vaccination.

Response #6: We made two changes to address the concerns raised here: (1) we required patients to have received their last required vaccine dose 14 days prior to COVID-19 infection to be considered fully vaccinated, and (2) we added a subanalysis that includes indicators for the amount of time separating a patient's most recent vaccination and COVID-19 infection. The first change results in subtly different cohorts (some people originally classified as fully vaccinated are now considered partially vaccinated and are excluded), so all tables and figures have been updated. Please see the following update to the exposure definition:

"Those who completed their vaccine regimen (two doses of BNT162b2 or mRNA-1273 or a single dose of Ad26.COV2.S) two weeks prior to index were considered vaccinated, while those with no recorded vaccines at index were considered unvaccinated."

And in the Results:

"The subanalysis did not offer robust evidence that the association between vaccination and long COVID diagnosis is dependent on the time between vaccination and acute COVID-19 onset. The full tables of subanalysis coefficients, including for indicators of vaccination timing, are shown in eTables 7–10."

And in the Discussion:

"Several studies conclude that the protective effect of vaccination on acute COVID-19 infection severity wanes over time (Andres, Meni), but we are unaware of any studies making the same claim for long COVID. As can be seen in eTables 7–10, the subanalysis incorporating time between vaccination and acute COVID-19 does not offer any evidence that the association between vaccination and long COVID diagnosis changes over time. The reference level for *Weeks Since Last Vaccination* is those who received their last vaccine dose at least 25 weeks prior to their COVID-19 infection. Negative coefficients for the modeled indicators suggest stronger protective associations. Three models present statistical significance ($\alpha=0.05$) for at least one indicator, indicating a significant difference between that level and those vaccinated 25+ weeks prior to COVID-19, but results are not consistent between models. Contrary to intuition and previously reported results with acute COVID-19, those vaccinated at least 25 weeks prior to COVID-19 are among the least likely to be diagnosed with long COVID across the four models. We do not present this as evidence that the benefits of vaccination with respect to long COVID do not wane. Caution should be used when interpreting conditional coefficients and investigating the time between vaccination and COVID-19 was not a primary focus in this study (Schisterman)."

Comment #7: Did the authors conduct a sub analysis considering separately those who received 2 doses vs 3 doses+? I am not sure why the authors consider one viral vector vaccine to be a completed course- while this may be the original recommendation for the J&J vaccine, this is not the case for the AZ vaccine. Patients who received one dose of an AZ vaccine should not be considered fully vaccinated.

Response #7: We did not conduct a sub analysis considering separately those who received 2 vs 3+ doses. Our cohort does not include any patients with a recorded AZ vaccine; we have clarified our exposure definition:

“Those who completed their vaccine regimen (two doses of BNT162b2 or mRNA-1273 or a single dose of Ad26.COV2.S) two weeks prior to index were considered vaccinated, while those with no recorded vaccines at index were considered unvaccinated.”

Reviewer #3

In this paper, the authors seek to estimate the association between pre-COVID vaccination and long COVID incidence. As the authors mentioned in their introduction, this question has been addressed elsewhere (e.g. in Al Aly et al 2022 and Taquet et al 2022). However, the question is far from answered in any definitive way and the present study has one unique feature: it is based on sites wherein recorded vaccination rates is close (> 66%) to what is expected.

It also has some important limitations which can hopefully be addressed as sensitivity analyses or further discussion.

Comment #1: The time of the index event (i.e. the month of the COVID-19 infection) is an important covariate and while it is “adjusted for” within the IPTW, it would be important to also run an analysis stratified by month of index event. This is because the time of the index event does not just affect the propensity of exposure, but also the duration of follow-up (and hence the probability of being censored).

Response #1: We agree with the need to adjust for the duration of follow-up depending on the time of the index event. The proportional hazards models will do this intrinsically, as individuals with a later index date will be censored sooner. We also include the month of COVID-19 infection as indicators in the logistic regression models for the outcome (in addition to the treatment weight models for IPTW), which allows the logistic regression models to adjust the baseline rate depending on the length of available follow-up. Ultimately, we are chiefly relying on the proportional hazard models to account for this effect; we believe that a stratified analysis using logistic regression models would add significant complexity to address the same concern.

Comment #2: The lack of adjustment for psychiatric comorbidities/history thereof is puzzling, given that (a) psychiatric symptoms are often considered to be part of long-COVID, and (b) psychiatric comorbidities affect patients’ healthcare utilization. This should be incorporated in the analysis.

Response #2: We agree that psychiatric comorbidities are important to consider as part of long COVID research in general. Patients in our study’s long COVID group may indeed have psychiatric comorbidities, which may have been taken into account by the physicians who assigned U09.9 or B94.8 diagnosis codes to those patients, or may have been picked up by the machine learning model when assigning a score. Thus, psychiatric comorbidities are accounted for in our work as a part of the overall constellation of factors exacerbating or resulting from long COVID. However, electronic health records rarely include a complete accounting of psychiatric conditions, which may be deliberately under-documented, or may be missing due to many patients seeking mental health care at private clinics. Due to this unevenness and high rate of missingness, adjusting for these factors in our analysis would likely introduce more bias. We believe they are appropriately accounted for without such an adjustment.

Comment #3: The inclusion of “visit to a long COVID clinic” in the primary outcome (clinical definition) affects the validity of that outcome as patients might attend long COVID clinic only to be told they do not have long COVID. It is unclear which individual outcome (U09.9, B94.8, or visit to an LC clinic) drive the incidence and hazard ratios for the primary composite outcome. Could the authors (a) report the total number of each of the three clinical events observed in their cohorts, and (b) report the results of a sensitivity analysis in which each event of the composite outcome is assessed individually?

Response #3: Thank you for pointing out this concern. We have added a table to the supplement summarizing the number of individuals with each of the three outcomes (some individuals appear in multiple rows if they received multiple distinct diagnoses):

Long COVID Diagnosis	Count of Unique Individuals
U09.9 Diagnosis Code	661
B94.8 Diagnosis Code	79
Visit to a Long COVID Clinic	91

It is clear that U09.9 is by far the most common diagnosis. The other diagnoses do not have sufficient sample sizes for us to feel confident in using them for independent analyses. We use long COVID clinic visits from only one partner facility, which requires a physician referral for admission into the clinic. While it is possible there are some misdiagnoses related to that outcome, we do not think it is likely to be higher than any other metric.

Comment #4: The actual incidence of long COVID is unknown. But of all incidences reported, the one reported here (1.5%) is among the lowest. This is not surprising given that many clinicians are probably not yet using the U09.9 code. One could therefore consider that the computational phenotypes with lower thresholds might better capture the proportion of patients with long COVID. The fact that little effect of vaccination is seen when the CP threshold is decreased calls for more cautious conclusion in the abstract (hence the need to rephrase the last sentence).

Response #4: We agree that this limitation calls for caution. Please refer to our response to reviewer #2’s comment #3 for a more detailed response on the study’s low long COVID prevalence. While a lower computational phenotype threshold better aligns with previously reported prevalences of long COVID, it also has a higher rate of misclassification likely to conservatively bias the measured associations with vaccination. Per your suggestion, we have added some qualifiers to the final sentence of the abstract, which now reads as follows.

“We found that vaccination was consistently associated with lower odds of long COVID clinical diagnosis and high-confidence computationally derived diagnosis after adjusting for sex, demographics, and medical history”

We also added to the paragraph in the discussion section in which we discuss the sensitivity analysis:

“In fact, our sample’s long COVID incidence rate at lower thresholds is closer to long COVID incidence rates reported elsewhere (although the true incidence rate of long COVID is unknown). This suggests a hypothesis that vaccination may be more effective at preventing clinically diagnosed long COVID than undiagnosed long COVID. More research is needed to determine the differences in high confidence and clinically diagnosed long COVID cases compared to low confidence and undiagnosed cases. If they are more severe, then our results could suggest that vaccination is associated with reduced severity of long COVID symptoms.”

Comment #5: In the primary analysis, patients had to have had a healthcare visit between 120 and 300 days after index. This implies that the results are not valid for the patients not making one such visit (which, according to Fig 1 might constitute the majority of patients). Could the authors run a sensitivity analysis in which all patients are included and in which they can be censored at their last visit (even if this occurs before Day 120) or censored at day 300 if their last visit was made over 300 days after the index?

Response #5: We agree this is a limitation, and we have added an additional sensitivity analysis to address it. Please see the following additions:

In the Methods:

“The fifth sensitivity analysis censored individuals after their last recorded healthcare visit. It also relaxed the requirement of having a visit at least 120 days after COVID-19 infection to at least 45 days after COVID-19 infection (when they became eligible for our outcome definition). The fifth analysis was only relevant for the proportional hazards model in the clinic-based analysis, as the computable phenotype model requires a post-COVID-19 visit at least 60 days after COVID-19 and censoring is not available in logistic regression models (Pfaff). Vaccination propensity scores as well as the model coefficients were recalculated for each of the modified cohorts in the fourth and fifth sensitivity analyses.”

In the Results:

“The results of the final two sensitivity analyses are summarized in eTable 6... Censoring patients in the clinic-based analysis after their last recorded healthcare visit and relaxing the requirement for a post-COVID-19 visit (from at least 120 days after to at least 45 days after) resulted in an association similar to our primary results.”

As well as this section in the Discussion:

“A sensitivity analysis that censors individuals after their last healthcare visit (rather than at the end of the study period) yields an association similar to our four primary results (eTable 6). Censoring is not possible in logistic regression models, but allows the proportional hazards model to relax the

assumption that individuals that established care at a facility continue to use that facility after their last recorded visit. For this analysis we required a visit at least 45 days after acute infection (instead of 120 days), as individuals with no visits at least 45 days after acute COVID-19 would be censored prior to becoming eligible for our outcome.”

Comment #6: Death was a censoring condition in the primary analysis. This raises the possibility of survivorship bias as vaccinated individuals are less likely to die from COVID. Could the authors run a sensitivity analysis in which death is included in a composite endpoint with long COVID to assess its influence on the primary results?

Response #6: The requirement for at least one healthcare visit at least 120 days after COVID onset ensures that patients must survive their acute COVID-19 event to become eligible for the study. As a result, we believe death among those in our study cohorts is more likely to be due to reasons unrelated to COVID-19 and may not be appropriate to be included as a composite endpoint with LC. Furthermore, the death rate in our cohort was very low (~0.1%).

Comment #7: One limitation of the use of a clinical diagnosis or computational phenotype is that it does not discern between the different symptoms of long-COVID. Both studies by Al Aly et al, and Taquet et al showed that the ‘protective effect’ of prior vaccination varied between symptoms. This should be better highlighted in the discussion.

Response #7: We agree that this is an important limitation. We have added the following paragraph to the discussion section.

“Our use of long COVID diagnosis and a computable phenotype as outcomes differentiate this study from others (Al-Aly, Tarquet, Simon), which measure the association between vaccination and a curated list of long COVID symptoms. Each approach has its strengths. Our clinical outcomes reduce measurement error due to false positives (e.g., long COVID symptoms caused by something other than long COVID). However, other studies show that long COVID symptoms differ in their relationship with vaccination. Our outcomes obscure such variation. We conclude that it is beneficial to study this relationship from both perspectives.”

REVIEWER COMMENTS

Reviewer #1 (Remarks to the Author):

The authors have addressed my comments.

Reviewer #2 (Remarks to the Author):

This paper uses a large database of electronic health records (EHR) to ascertain an association between vaccination status and post-acute COVID symptoms. The main added value of this study is the use of such a database as opposed to self reported symptoms. Both methodologies have their own intrinsic limitations and it is reassuring that the findings of this study are consistent with the existing evidence, strengthening the claim that vaccination mitigates long COVID symptoms. In this revised version, the authors have taken into consideration all the reviewers' comments and modified the manuscript accordingly. The manuscript is methodologically sound and acknowledges the limitations of the dataset the authors use, in particular in terms of classification of exposure and outcome. The authors have conducted several sensitivity analyses to try and mitigate these limitations, and offer an interpretation that takes these limitations into account, specifically that because of potential misclassification bias the strength of association is biased towards the null. I therefore believe that this study in its current state can be published and will strengthen the emerging body of evidence that exists around vaccination and long COVID

Reviewer #3 (Remarks to the Author):

The authors provided partial responses to my initial comments, and some of those comments remain unanswered which means that I am not yet convinced by the robustness of the conclusions. I believe that addressing them will only strengthen the paper, even if they lead to more nuanced conclusions.

1) My second comment pertained to the confounding of association by psychiatric comorbidities/history at baseline. Adjusting for psychiatric comorbidities would be simple and yet the authors decided to rebut rather than address this comment, stating that "electronic health records rarely include a complete accounting of psychiatric conditions, which may be deliberately under-documented, or may be missing due to many patients seeking mental health care at private clinics". This comment by the authors could be applied to many comorbidities (not just psychiatric ones) and is a limitation of electronic health records data. There is no reason to believe that such under-reporting of psychiatric disorders would disproportionately affect one cohort versus the other.

I therefore urge the authors to adjust for psychiatric comorbidities by including diagnostic codes as separate comorbidities. At a minimum, I would expect them to adjust for each of the main subcategories of psychiatric disorders: F10-F19, F20-F29, F30-F39, F40-F48.

2) My third comment pertained to an analysis of separate outcomes to assess whether any of them are driving the results. The breakdown by outcome subcategories is informative, and I understand why the authors fear that analyzing B94.8 or Visit to a Long COVID Clinic on their own would be underpowered. However, they can still run a sensitivity analysis in which only code U09.9 (i.e. the most prevalent) is used as an outcome. This is particularly important since the authors mentioned that visits to long COVID clinic were only available for one site.

3) Regarding my 4th comment, I think the authors appropriately addressed it in the discussion, but the abstract could be clearer (as it currently still gives the impression that results were consistent throughout the spectrum of long-COVID assessments). I suggest the following:

“We found that vaccination was consistently associated with lower odds of both long COVID clinical diagnosis and highly specific computationally derived diagnosis (but not with lower odds of highly-sensitive computational diagnoses) after adjusting for sex, demographics, and medical history”

4) Regarding my 5th comment, the sensitivity analysis provided by the authors still requires that participants made at least one visit after their COVID-19 diagnosis. It is quite possible (if not likely) that a large proportion of those who made no visit at all were actually quite well after COVID-19. This means that the study results are still potentially prone to bias if, for instance, many of those who were not vaccinated prior to their COVID-19 diagnosis made no visit to a healthcare facility after COVID-19. Because all those who made no visit within the first 45 days are excluded (rather than censored) from the primary and sensitivity analyzes, it is impossible to know whether this is impacting the results.

Could the authors run the sensitivity analysis suggested in my initial comment, in which all participants are included and censored on Day 1 of follow-up if they made no visit at all (this is very different from excluding them). The fact that “they would be censored before becoming eligible for outcomes” is not an issue.

Reviewer #3

The authors provided partial responses to my initial comments, and some of those comments remain unanswered which means that I am not yet convinced by the robustness of the conclusions. I believe that addressing them will only strengthen the paper, even if they lead to more nuanced conclusions.

Comment #1: My second comment pertained to the confounding of association by psychiatric comorbidities/history at baseline. Adjusting for psychiatric comorbidities would be simple and yet the authors decided to rebut rather than address this comment, stating that “electronic health records rarely include a complete accounting of psychiatric conditions, which may be deliberately under-documented, or may be missing due to many patients seeking mental health care at private clinics”. This comment by the authors could be applied to many comorbidities (not just psychiatric ones) and is a limitation of electronic health records data. There is no reason to believe that such under-reporting of psychiatric disorders would disproportionately affect one cohort versus the other.

I therefore urge the authors to adjust for psychiatric comorbidities by including diagnostic codes as separate comorbidities. At a minimum, I would expect them to adjust for each of the main subcategories of psychiatric disorders: F10-F19, F20-F29, F30-F39, F40-F48.

Response:

We appreciate the comment, and initially reran the analysis including an indicator derived from the VSAC value set for Mental Behavioral and Neurodevelopmental Disorders (OID: 2.16.840.1.113883.3.464.1003.105.11.1207). The largest impact on the treatment effect across any of the four primary models was an absolute change in the vaccination coefficient of 0.01, barely discernible at the reported precision. However, we received critical feedback from the project’s patient advocate on the inclusion of this variable. Some researchers are framing long COVID as a primarily psychiatric condition (we are not implying that the reviewer is among them), downplaying the considerable physiological symptoms experienced by those with long COVID. We feel strongly that, were we to include psychiatric comorbidities in the model, they would need to be accompanied by considerable discussion to justify their inclusion. We feel this discussion would distract from the focus of the study: the association between pre-COVID-19 vaccination and long COVID diagnosis. Given that we observed no impact on the treatment effect after introducing a psychiatric comorbidity indicator, and the concerns raised by our patient advocate, we have elected to not include this variable in the reported results.

Comment #2: My third comment pertained to an analysis of separate outcomes to assess whether any of them are driving the results. The breakdown by outcome subcategories is informative, and I understand why the authors fear that analyzing B94.8 or Visit to a Long COVID Clinic on their own would be underpowered. However, they can still run a sensitivity analysis in which only code U09.9 (i.e. the

most prevalent) is used as an outcome. This is particularly important since the authors mentioned that visits to long COVID clinic were only available for one site.

Response:

We have added a sensitivity analysis that exclusively uses the U09.9 ICD-10 code to label long COVID. It is described in the Results:

“In the clinic-based cohort, an analysis showed the results to be robust to using only the most specific ICD-10 code (U09.9) to label long COVID”

And Discussion:

“We accepted B94.8 as a long COVID diagnosis because use of the code in our data by mid-2021 was 40 times higher than its baseline use in 2018 and 2019. A sensitivity analysis using only U09.9 returned nearly identical results.”

And Methods:

“They test the sensitivity of the IPTW-adjusted and unadjusted vaccination status coefficients in the logistic regression and proportional hazards models across four dimensions: (1) CP score threshold (0.3 to 0.95), (2) with or without independent features in addition to vaccination, (3) including or not including post-index vaccinations as a censoring event, and (4) using only U09.9 diagnoses to label long COVID. The first sensitivity dimension was not relevant for the clinic-based outcome, the third was not relevant for logistic regression analyses, and the fourth was not relevant for the model-based outcome.”

Comment #3: Regarding my 4th comment, I think the authors appropriately addressed it in the discussion, but the abstract could be clearer (as it currently still gives the impression that results were consistent throughout the spectrum of long-COVID assessments). I suggest the following:

“We found that vaccination was consistently associated with lower odds of both long COVID clinical diagnosis and highly specific computationally derived diagnosis (but not with lower odds of highly-sensitive computational diagnoses) after adjusting for sex, demographics, and medical history”

Response:

We agree that a clarification that the association between long COVID and vaccination is weaker among less confident computational diagnoses is reasonable. However, in keeping with the abstract’s word limit, we do not feel that enough text could be removed from the abstract to accommodate the clarification without introducing some confusion. In its current form, we believe that the abstract’s reference to “high-confidence computationally derived diagnosis” is clear and not misleading, and the dialogue in the Discussion regarding less confident diagnoses is sufficient to address this concern.

Comment #4: Regarding my 5th comment, the sensitivity analysis provided by the authors still requires that participants made at least one visit after their COVID-19 diagnosis. It is quite possible (if not likely) that a large proportion of those who made no visit at all were actually quite well after COVID-19. This means that the study results are still potentially prone to bias if, for instance, many of those who were not vaccinated prior to their COVID-19 diagnosis made no visit to a healthcare facility after COVID-19. Because all those who made no visit within the first 45 days are excluded (rather than censored) from the primary and sensitivity analyzes, it is impossible to know whether this is impacting the results.

Could the authors run the sensitivity analysis suggested in my initial comment, in which all participants are included and censored on Day 1 of follow-up if they made no visit at all (this is very different from excluding them). The fact that “they would be censored before becoming eligible for outcomes” is not an issue.

Response:

We have refactored the censoring sensitivity analysis as suggested. In particular, this new analysis makes the following changes from the primary survival analysis:

- Only the clinic-based cohort is used, as the computable phenotype requires a post-index visit
- There is no requirement for any visits after index
- Day 0 of the survival analysis is day of index, rather than 45 days post-index
- Individuals are censored as of their last visit, even if that visit occurs before they become eligible for long COVID (45 days after index). Those with no post-index visits are censored on Day 1.

It is described in the Results:

“Censoring patients in the clinic-based analysis after their last recorded healthcare visit and eliminating the requirement for a post-COVID-19 visit resulted in a slightly stronger, but not significantly different, association as in our primary results.”

And Discussion:

“A sensitivity analysis that censors individuals after their last healthcare visit (rather than at the end of the study period) yields an association similar to our four primary results (eTable 6). Censoring is not possible in logistic regression models, but allows the proportional hazards model to relax the assumption that individuals that established care at a facility continue to use that facility after their last recorded visit. For this analysis we did not require a recorded visit after the acute COVID-19 infection, but individuals remained ineligible for long COVID designation until 45 days after infection.”

And Methods:

“The sixth sensitivity analysis eliminated the requirement for a recorded healthcare visit after COVID-19 infection and censored individuals after their last visit. Those without a visit after COVID-19 were censored the day after index. For this analysis, the modeled time period began the day after index

(instead of 45 days after index in the primary analysis), though individuals remained ineligible for long COVID designation until 45 days after index. The sixth analysis was only relevant for the proportional hazards model in the clinic-based analysis, as the computable phenotype model requires a post-COVID-19 visit at least 60 days after COVID-19 and censoring is not available in logistic regression models.”